# Presenilin 2 N141I mutation induces hyperactive immune response through the epigenetic repression of REV-ERBα

Hyeri Nam[1], Younghwan Lee[1], Boil Kim[1], Ji-Won Lee [1], Seohyeon Hwang [1], Hyun-Kyu An[1], Kyung Min Chung[1], Youngjin Park[1], Jihyun Hong[1], Kyungjin Kim[1], Eun-Kyoung Kim[1,2], Han Kyoung Choe[1] & Seong-Woon Yu [1✉]

Hyperimmunity drives the development of Alzheimer disease (AD). The immune system is under the circadian control, and circadian abnormalities aggravate AD progress. Here, we investigate how an AD-linked mutation deregulates expression of circadian genes and induces cognitive decline using the knock-in (KI) mice heterozygous for presenilin 2 N141I mutation. This mutation causes selective overproduction of clock gene-controlled cytokines through the DNA hypermethylation-mediated repression of REV-ERBα in innate immune cells. The KI/+ mice are vulnerable to otherwise innocuous, mild immune challenges. The antipsychotic chlorpromazine restores the REV-ERBα level by normalizing DNA methylation through the inhibition of PI3K/AKT1 pathway, and prevents the overexcitation of innate immune cells and cognitive decline in KI/+ mice. These results highlight a pathogenic link between this AD mutation and immune cell overactivation through the epigenetic suppression of REV-ERBα.

[1] Department of Brain and Cognitive Sciences, Daegu Gyeongbuk Institute of Science and Technology (DGIST), Daegu 42988, Republic of Korea. [2] Neurometabolomics Research Center, DGIST, Daegu 42988, Republic of Korea. ✉email: yusw@dgist.ac.kr

Alzheimer disease (AD) is the most common neurological disorder characterized by progressive loss of cognitive functions. The key dual pathological hallmarks of AD are extracellular amyloid plaques and intracellular neurofibrillary tangles[1]. With the increase in lifespan, AD poses a severe burden on global public health. Most cases of AD are sporadic, but a small percentage is inherited and is caused by the mutated versions of the genes including *APP*, *PSEN1*, and *PSEN2*, which encode amyloid precursor protein, presenilin 1, and presenilin 2, respectively[2]. Identification of amyloid β (Aβ) peptide in the senile plaques and widespread occurrence of amyloid deposition have led to the prevailing amyloid cascade hypothesis[3]. Genetic studies also suggested that familial AD (FAD) mutations in the above genes elevate amyloidogenic Aβ42 production[4]. However, intensive efforts over the last several decades aimed at clearing or preventing amyloid deposits have failed to prove the central role of Aβ production and amyloid deposition in the pathogenesis of AD[5].

There is increasing recognition that innate immune response contributes to AD pathogenesis[6]. Systemic inflammation markers are strongly associated with an increased likelihood of dementia[7], and high levels of inflammatory mediators including interleukin-6 (IL-6) are found in the plasma of AD patients[8]. Cognitive status in patients with AD is inversely correlated with microglial activation but not Aβ load[9,10]. Furthermore, genome-wide association studies have clearly implicated innate immunity and inflammation in the pathogenesis of AD[11,12]. Therefore, it is now widely appreciated that inflammation is not only a central epiphenomenon in AD, but represents a major risk factor for the pathogenesis of AD[7].

Innate immunity is under circadian control, and circadian proteins can directly control the expression of genes involved in the innate immune system[13,14]. Circadian rhythm is an endogenous biological rhythm with an approximately 24-h period and regulated by interlocked transcription–translation feedback loop system[15,16]. REV-ERBα (encoded by the *NR1D1* gene) is a core inhibitory component of this circadian system and functions as a transcriptional repressor in the auxiliary loop by suppressing the expression of BMAL1 (encoded by the *ARNTL* gene)[17].

Interestingly, recent studies have revealed an intimate link between circadian abnormalities and AD[18–25]. In the brain, molecular clock machineries are found not only in neurons, but also in glia, suggesting the importance of a functional molecular clockwork for proper glial function[26,27]. Therefore, it is imperative to understand the functional roles of the immune cell clock genes in the progress of AD from the aspect of hyperimmunity.

PSEN has been implicated in inflammation[28,29]. Especially, PSEN2 seems to be the main PSEN in microglia[29]. However, how the *PSEN2* FAD mutations affect the innate immune system remains largely unknown. Therefore, mouse models mimicking human *PSEN2* mutations would be desirable to delineate the molecular pathways underlying hyperimmunity-driven AD pathogenesis and cognitive impairment. The Asn residue at position 141 in PSEN2 is conserved between human and mouse, and its substitution by Ile (N141I) is tightly linked to a form of early-onset FAD with strong penetrance[30].

Here, we generated knock-in (KI) mice expressing *Psen2* with the N141I mutation and provide evidence that mice heterozygous for *Psen2* N141I are vulnerable to otherwise innocuous, mild inflammatory challenges and exhibit a hyperactive immune response and memory deficits because of selective overproduction of clock gene-controlled cytokines. This hyperimmunity is due to the DNA methylation-mediated epigenetic suppression of the clock protein REV-ERBα. Therefore, we identify epigenetic repression of REV-ERBα as a missing link between the *Psen2* N141I FAD mutation and hyperimmune activity.

## Results

**$Psen2^{N141I/+}$ mice overproduce clock gene-controlled cytokines in response to LPS.** To investigate the pathogenic function of the *Psen2* N141I mutation in vivo, we generated KI mice harboring the *Psen2* N141I allele ($Psen2^{N141I/+}$ and $Psen2^{N141I/N141I}$) (Fig. 1a). The substitution of N to I (AAC to ATC) was confirmed by genomic sequencing (Fig. 1b). Human AD patients with FAD mutations are heterozygous[30]. Therefore, to more accurately recapitulate human AD and maintain the endogenous expression level, we used heterozygous $Psen2^{N141I/+}$ (KI/+) mice for all experiments. The protein and mRNA levels of PSEN2 in KI/+ primary microglia and bone marrow-derived macrophages (BMDM) were comparable to those in WT cells (Supplementary Fig. 1a, b), indicating that the N141I mutation did not affect PSEN2 expression. Also, this mutation did not induce a compensatory change in PSEN1 expression (Supplementary Fig. 1a, b). We estimated γ-secretase activity on the basis of the cleavage of N-cadherin and mRNA expression level of *Hes5* as a downstream target of the Notch pathway[31], and found no difference between N141I KI/+ and WT genotypes in microglia and BMDM (Supplementary Fig. 1c, d).

To explore the functional consequences of the *Psen2* N141I mutation in innate immunity, we examined whether this mutation affects inflammatory responses in $Psen2^{N141I/+}$ mice following systemic injection of lipopolysaccharide (LPS) derived from *Escherichia coli* O111:B4. In mice, the immune response peaks in the hours around the beginning of the active phase[27]. Therefore, we injected WT and KI/+ mice intraperitoneally (i.p.) with various doses of LPS at 6:00 PM with light-on at 7:00 AM (Fig. 1c) and monitored the inflammatory response after 20 h. Compared with WT mice, KI/+ mice exhibited a higher circulating level of IL-6 at all LPS doses tested, and the relative difference between the genotypes was more pronounced at the lower doses (Fig. 1d). On the other hand, the blood levels of TNF-α were the same in both genotypes at all doses (Fig. 1e). The lowest dose (0.35 μg/kg body weight) we tested did not induce TNF-α secretion (Fig. 1f) in either WT or KI/+ mice, but still increased the blood level of IL-6 in KI/+ mice (Fig. 1f).

To explore the mechanisms underlying the overactive response of IL-6 by *Psen2* N141I mutation, we compared the profiles of secreted cytokines using enzyme-linked immunosorbent assay (ELISA)-based cytokine protein array between WT and KI/+ microglia after LPS treatment. Among 40 targets, we detected 13 cytokines and observed increased secretion of C-X-C motif chemokine ligand 1 (CXCL1), C-C motif chemokine ligand 2 (CCL2), and CCL5 in addition to IL-6 in KI/+ microglia (Fig. 2a, b). Furthermore, their transcript levels were significantly upregulated in KI/+ microglia and BMDM relative to WT (Fig. 2c, d). To examine whether this set of cytokines is also more produced in KI/+ mice than WT, we measure their blood levels in mice injected with the dose of LPS that induced only IL-6 but not TNF-α secretion, and observed the same hyper response of this set of cytokines (Fig. 2e).

On the other hand, the secretion of CCL3, CCL4, and CXCL2 was lower in KI/+ microglia than WT by LPS treatment (Fig. 2b), but these cytokines showed no difference in blood serum levels between WT and KI/+ mice following LPS injection (Fig. 2f). Therefore, reduced secretion of these cytokines seems limited to in vitro microglia culture and subject to additional regulation in vivo.

Interestingly, all four increased cytokines (IL-6, CXCL1, CCL2, and CCL5) are known to be regulated by the circadian rhythm[32]. In contrast, secretion of TNF-α and its mRNA level were changed by LPS treatment to the same extent between WT and KI/+ genotypes in both microglia and BMDM (Fig. 2b–d). IL-1β is another major cytokine and is released through the inflammasome complex, such as the NLRP3 inflammasome in innate

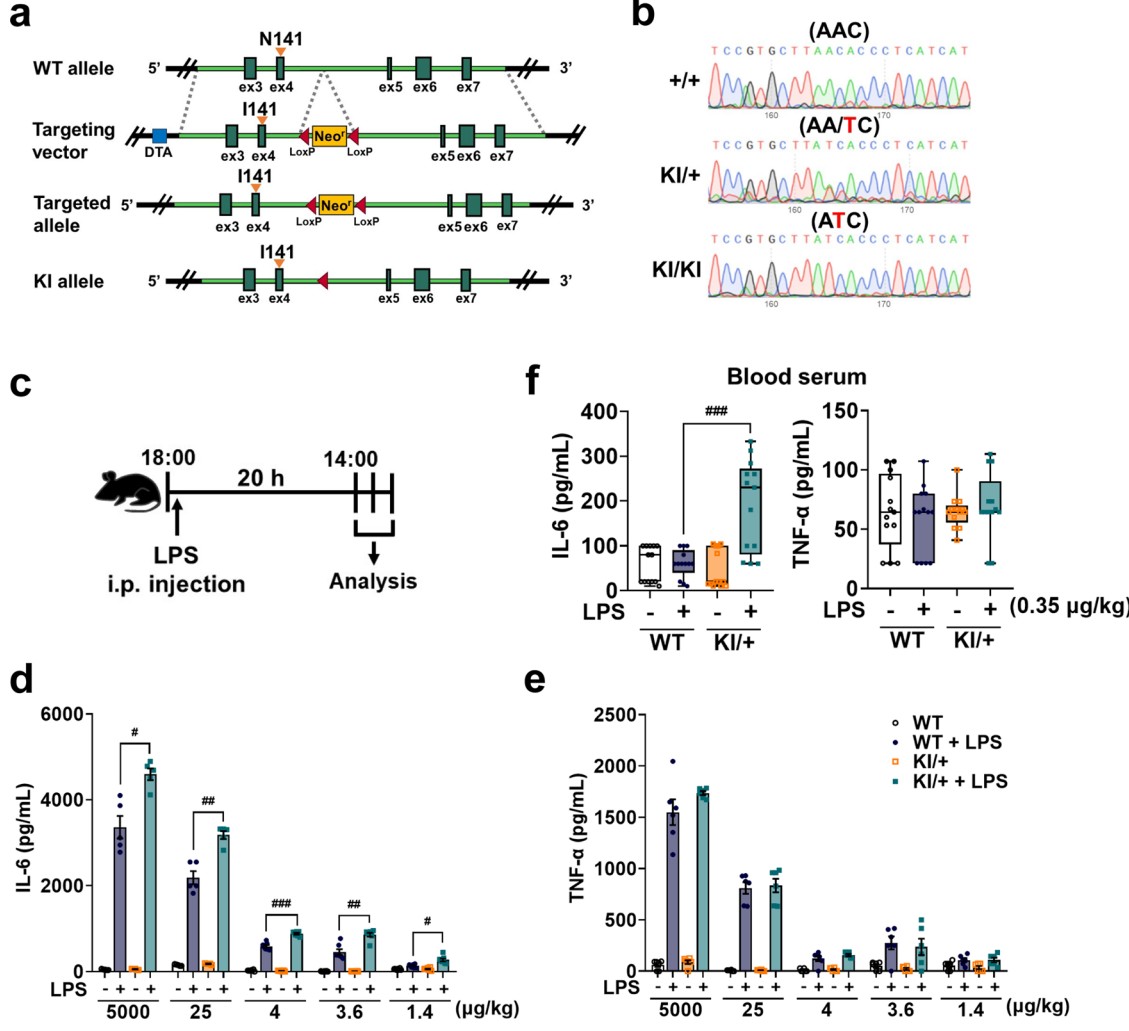

**Fig. 1 LPS injection elicits overactive response of IL-6 but not TNF-α in *Psen2*^N141I/+ mice. a** Strategy for targeted insertion of the N141I point mutation into the murine *Psen2* locus to generate a KI mouse model. **b** Sanger sequencing chromatograms of genomic DNA from wild type (WT), *Psen2*^N141I/+ (KI/+), and *Psen2*^N141I/N141I (KI/KI) mice. **c** Schematic illustration of the experimental schedule. **d, e** ELISA of IL-6 (**d**) (n = 5 mice for 5000 and 25 μg/kg LPS-injection group; n = 7 mice for 4, 3.6, and 1.4 μg/kg LPS-injection group) (LPS-injected WT vs LPS-injected KI/+: #p = 0.0343, ##p = 0.0061, ###p < 0.0001, ##p = 0.0021, and #p = 0.0322; two-way ANOVA) and TNF-α (**e**) (n = 6 mice per group) in blood serum from WT and KI/+ mice (8-week-old, male) i.p. injected with various doses of LPS. **f** ELISA of IL-6 and TNF-α in blood serum from WT and KI/+ mice (8-week-old, male, n = 13 mice per group) 20 h after i.p. injection of LPS (0.35 μg/kg) (LPS-injected WT vs LPS-injected KI/+: ###p < 0.0001; one-way ANOVA). The bounds of the box represent 25th to 75th percentiles ranges. The center lines with the box represent the mean value and whiskers represent the minimum to the maximum range. Data are mean ± SEM. #p < 0.05, ##p < 0.01, and ###p < 0.001 for indicated comparisons. Source data are provided as a Source Data file.

immune cells[33]. We activated the NLRP3 inflammasome by stimulating LPS-primed microglia with nigericin, as in our previous study[33], and found a similar amount of released IL-1β and its transcript levels between WT and KI/+ microglia (Supplementary Fig. 2). Of note, the expression of both *Tnf* and *Il-1b* is circadian cycle–independent[32]. To sum up, the *Psen2* N141I mutation induced hyperactive responses of clock gene-controlled cytokines in microglia and BMDM, innate immune cells representative of the brain and periphery, respectively.

***Psen2*^N141I/+ mice display proinflammatory microglial morphology and suffer memory deficit in response to the low dose of LPS ineffective in WT mice.** Microglia morphology is closely related to their function and microglial activation is characterized by cell shape change[34]. Therefore, we wondered whether the increased production of four cytokines is associated with changes in the morphology in KI/+ microglia. We examined microglia shapes in the hippocampus of KI/+ and WT mice by

immunohistochemical analyses with an antibody against a microglia-specific marker, IBA-1. In the WT mice, microglia had a small cell body with highly ramified processes; consistent with no induction of cytokine release, a low dose of LPS (0.35 μg/kg body weight) did not change their morphology (Fig. 3a). On the other hand, hippocampal microglia in KI/+ mice, even in the absence of LPS challenge, already had a round enlarged soma with shorter processes, and these morphological features were furthered by LPS injection at the dose ineffective in WT mice (Fig. 3a). We used the confocal images of microglia to reconstruct their 3D morphology and measured morphological parameters using IMARIS software (Fig. 3b). Total dendrite length and the number of dendrite terminal points of each microglial cell were lower in KI/+ than in WT mice and further reduced by LPS injection (Fig. 3c). Sholl analysis measures the average number of dendrite intersections on concentric rings spaced at 1-μm intervals from the center of the cell body. Sholl analysis revealed that the hippocampal microglia of *Psen2*^N141I/+ mice had fewer

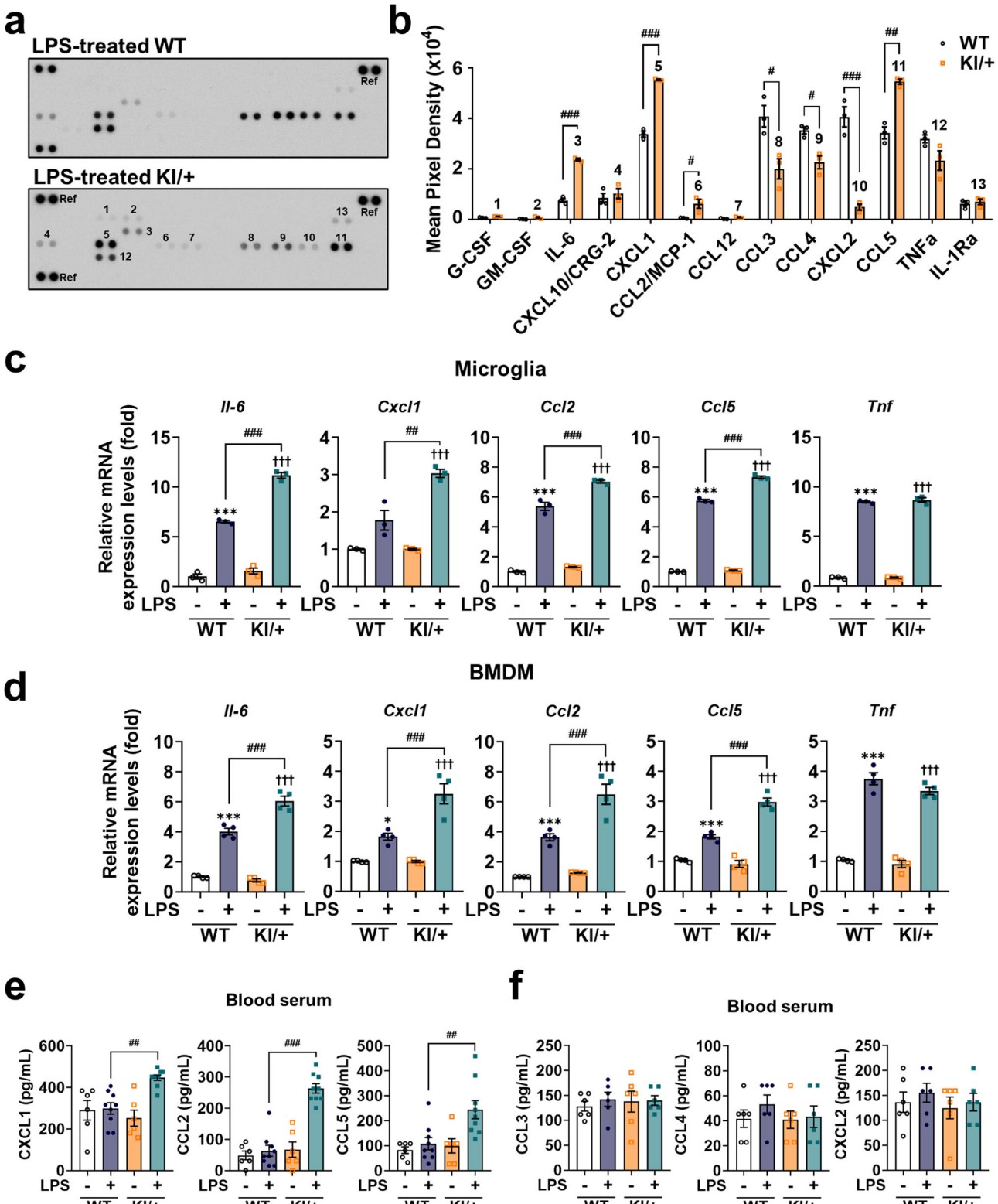

dendrite intersections than those of WT mice and exhibited dwindled morphology, and these characteristics became more distinct in response to LPS injection (Fig. 3c), consistent with microglial activation. Therefore, on the basis of morphology, microglial activation was evident in KI/+ mice at baseline and was further induced by mild LPS challenge.

To determine whether the differential increase in the production of a subset of inflammatory cytokines in KI/+ is reflected by

cognitive decline, we injected mice with the same low dose of LPS (0.35 μg/kg body weight), and conducted a Y-maze test 20 h after LPS injection. The arm alternation in the Y-maze was significantly decreased by LPS only in KI/+ mice (Fig. 3d), while the total number of arm entries was similar across all the groups, indicating normal locomotor function (Fig. 3e). To further examine learning and memory, we conducted a T-maze test with a food reward 20 h after LPS injection. Similar to the Y-maze test,

**Fig. 2 _Psen2_ N141I mutation increases production of clock gene-controlled cytokines following LPS treatment. a** Proteome ELISA of 40 cytokines in the culture media of WT and KI/+ microglia after LPS (1 μg/mL) treatment for 12 h. The blots shown are representative of three independent experiments with similar results. Signals were normalized to pixel density of indicated reference (Ref) spots. **b** Quantification of the average signal (pixel density) as relative changes in cytokine levels between WT and KI/+ samples ($n = 3$) (LPS-treated WT vs KI/+: ###$p < 0.0001$, ###$p < 0.0001$, #$p = 0.0351$, #$p = 0.0233$, #$p = 0.0128$, ###$p = 0.001$, and ##$p = 0.0013$; unpaired _t_-test). **c, d** Relative mRNA expression levels of the increased cytokines (_Il-6_, _Cxcl1_, _Ccl2_, and _Ccl5_) and _Tnf_ after LPS (1 μg/mL) treatment for 12 h in primary microglia (**c**; $n = 3$) (untreated WT vs LPS-treated WT: ***$p < 0.0001$, *$p = 0.0212$, ***$p < 0.0001$, ***$p < 0.0001$, and ***$p < 0.0001$; untreated KI/+ vs LPS-treated KI/+: †††$p < 0.0001$; LPS-treated WT vs LPS-treated KI/+: ###$p < 0.0001$, ##$p = 0.0012$, ###$p = 0.0002$, and ###$p < 0.0001$; one-way ANOVA) and BMDM (**d**; $n = 4$) (untreated WT vs LPS-treated WT: ***$p < 0.0001$, *$p = 0.0296$, ***$p = 0.0009$, ***$p = 0.0004$, and ***$p < 0.0001$; untreated KI/+ vs LPS-treated KI/+: †††$p < 0.0001$; LPS-treated WT vs LPS-treated KI/+: ###$p < 0.0001$, ###$p = 0.0005$, ###$p = 0.0005$, and ###$p < 0.0001$; one-way ANOVA). mRNA levels were normalized to _Actb_. **e, f** ELISA of CXCL1, CCL2, and CCL5 (**e**; 8-week-old, male, $n = 6$ mice for WT and KI/+; $n = 9$ mice for LPS-injected WT and KI/+) (LPS-injected WT vs LPS-injected KI/+: ##$p = 0.0048$, ###$p < 0.0001$, and ##$p = 0.0086$; one-way ANOVA) and CCL3, CCL4, and CXCL2 (**f**; 8-week-old, male, $n = 6$ mice) in blood serum from WT and KI/+ mice 20 h after i.p. injection of LPS (0.35 μg/kg). Data are mean ± SEM. *$p < 0.05$, and ***$p < 0.001$ vs untreated WT control. †††$p < 0.001$ vs untreated KI/+ control. #$p < 0.05$, ##$p < 0.01$, and ###$p < 0.001$ for the indicated comparisons. Source data are provided as a Source Data file.

the success rate for entering the correct arm was significantly lower in LPS-injected KI/+ mice than in the other groups (Fig. 3f, g).

KI/+ mice developed no sickness behavior nor did they differ from WT mice in spontaneous home-cage behaviors at this low dose of LPS, when monitored by LABORAS (Supplementary Fig. 3a, b). Therefore, there was no difference with respect to basal activity patterns between the genotypes. In sum, a low dose of LPS induced a hyperactive immune response and caused memory deficit through the overproduction of clock gene-controlled cytokines including IL-6 in _Psen2_N141I/+ mice, while the same dose of LPS was innocuous to WT mice.

**_Psen2_ N141I mutation induces secretion of clock gene-controlled cytokines and memory deficits in responses to fibrillar Aβ.** To establish a link between _Psen2_ N141I mutation and AD immunopathology, we examined whether synthetic Aβ also induces hyperactive immune responses in _Psen2_N141I/+ mice. Fibrillar form of Aβ42 (fAβ42) was injected intracerebroventricularly (i.c.v.) in the right lateral ventricle (Fig. 4a), according to the previous studies[35,36], and immune responses and memory deficits were analyzed 7 days later. The dosage of fAβ42 used in the current study did not elicit an immune response or memory impairment in WT mice (Fig. 4b, d). On the other hand, the arm alternation in the Y-maze was declined in fAβ42- injected KI/+ mice (Fig. 4b) with a similar total number of arm entries across all experimental groups (Fig. 4c). Also, increases in the circulating levels of clock gene-controlled cytokines including IL-6 were observed in fAβ42-injected KI/+ mice (Fig. 4d), while the TNF-α levels remained the same as a control in both genotypes (Fig. 4d).

In a similar manner to LPS treatment, fAβ42 treatment induced more production of clock gene-controlled cytokines (IL-6, CXCL1, CCL2, and CCL5) in KI/+ microglia than WT, while an increase in the TNF-α level was similar between two genotypes (Fig. 4e). Therefore, _Psen2_ N141I mutation with fAβ42 i.c.v. injection induces hyperactive immune responses and memory decline, thus explaining the link between hyperimmunity and AD pathology.

**_Psen2_ N141I mutation decreases the expression of clock genes, but innate immune cells remain rhythmic.** Because the _Psen2_ N141I mutation increased the production of clock gene-controlled cytokines, we examined whether there was an alteration in the expression levels of clock genes in WT and KI/+ microglia and BMDM without cell synchronization. The steady-state mRNA levels of _Nr1d1_, _Clock_, _Cry1_, _Per1_, and _Per2_ genes were significantly reduced and conversely the _Arntl_ (encoding BMAL1) level was increased in KI/+ microglia and BMDM (Fig. 5a, b). Among the altered genes, the oscillation of _Nr1d1_ was severely impaired and its amplitude was greatly decreased in KI/+

microglia (Fig. 5c). These results suggest that the _Psen2_ N141I mutation impairs the expression of microglial clock genes. To validate the decrease of _Nr1d1_ expression in KI/+ immune cells under more physiological conditions, we performed acute isolation of microglia and peritoneal macrophages from _Psen2_N141I/+ mice and WT controls, and confirmed a significant decrease in the mRNA levels of _Nr1d1_ in KI/+ cells (Fig. 5d). Thus, compared with WT, _Nr1d1_ expression is suppressed in _Psen2_ N141I KI/+ immune cells.

To compare the robustness of daily oscillations in microglia and BMDM between the WT and KI/+ genotypes, we monitored circadian clock oscillations in primary microglia and BMDM derived from _Per2::Luc_ mice crossed with KI/+ (_Per2::Luc;Psen2_N141I/+) or WT littermates (_Per2::Luc;Psen2_+/+). _Per2::Luc_ reporter mice, in which the luciferase (_Luc_) gene is fused in-frame to the 3′ end of the endogenous mouse _Per2_ gene, show robust circadian rhythms in both SCN and peripheral tissues[37]. Since cell-autonomous circadian clock dampens gradually over time if it is not synchronized with the master SCN clock through neuronal signaling[38], this reporter line is a good system to compare and analyze the dynamics and robustness of cell-autonomous circadian oscillations by bioluminescence recording. WT microglia cultures exhibited robust luminescence oscillations for 3 days, whereas KI/+ microglia cultures displayed two robust circadian peaks followed by a barely recognizable third peak (Fig. 5e). Statistical analysis revealed that the amplitude was reduced in KI/+ microglia compared to WT, while the circadian period was similar between the two groups (Fig. 5e). BMDM also exhibited similar alterations in circadian oscillation, comparable period but decreased amplitude in KI/+ cells relative to WT (Fig. 5f). To examine the effects of _Psen2_ mutation on non-immune cells, we monitored the robustness of daily oscillation in mouse embryonic fibroblasts (MEF). Interestingly, KI/+ MEF exhibited the same circadian oscillation period and amplitude as those of WT cells with an equal expression level of REV-ERBα (Supplementary Fig. 4). Therefore, _Psen2_ N141I mutation seems to affect _Nr1d1_ level only in immune cells; nevertheless, immune cells derived from _Psen2_N141I/+ mice remain rhythmic. Since REV-ERBα can control the immune response and is known as a transcriptional suppressor of IL-6 induction[32,39,40], these data suggest that immunomodulatory but not circadian effects of REV-ERBα may contribute to the hyperimmunity phenotypes of KI/+ immune cells.

**Downregulation of REV-ERBα underlies selective overproduction of IL-6 in KI/+ microglia.** Consistent with the lower _Nr1d1_ mRNA level in KI/+ cells (Fig. 6a, b), protein levels of REV-ERBα were also much lower in KI/+ microglia and BMDM than in WT counterparts (Fig. 6a). Consistent with a previous

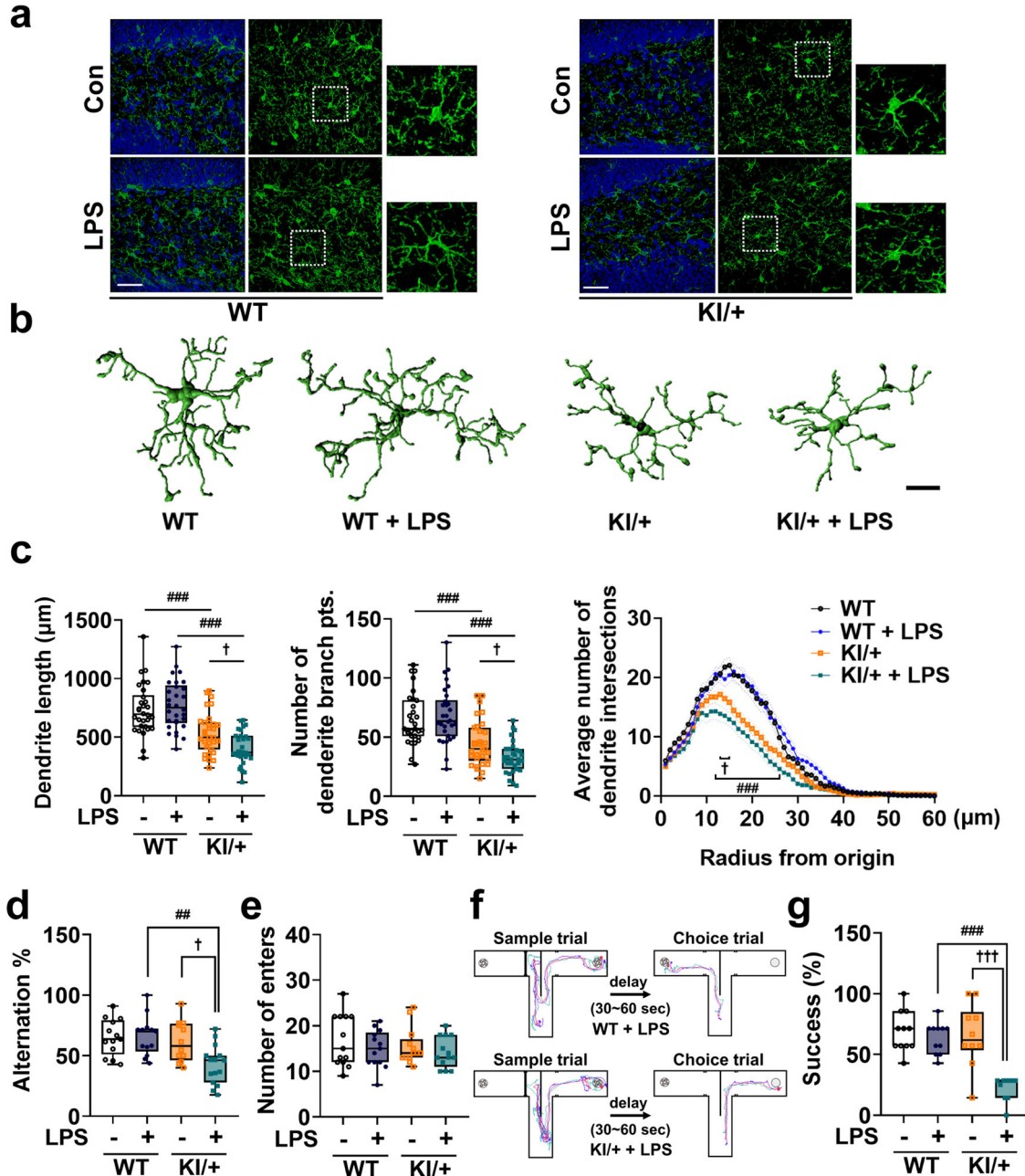

**Fig. 3 *Psen2* N141I mutation induces proinflammatory morphological changes in microglia and causes memory deficit in response to a mild LPS challenge. a** Representative immunofluorescent images by IBA-1 staining in hippocampus of WT and KI/+ mice (8-week-old, male, $n = 4$ mice with similar results) 20 h after i.p. injection of LPS (0.35 μg/kg body weight). Insets, 2.5-fold enlarged images. Scale bar, 25 μm. **b** Representative 3D filament tracking images of IBA-1 signals made by IMARIS software. Scale bar, 10 μm. **c** Dendrite length, number of branching branch points, and Sholl radius analysis. Data were extracted from FilamentTracker in IMARIS analysis ($n = 30$ cells per group from four mice) (WT vs KI/+: ###$p = 0.0002$ and ###$p = 0.0003$; KI/+ vs LPS-injected KI/+: †$p = 0.0448$ and †$p = 0.039$; LPS-injected WT vs LPS-injected KI/+: ###$p < 0.0001$; one-way ANOVA; radius 13–15 μm, KI/+ vs LPS-injected KI/+: †$p = 0.0113, 0.0362,$ and $0.0398$; radius 11–29 μm, LPS-injected WT vs LPS-injected KI/+: ###$p < 0.0001$; two-way ANOVA). **d**, **e** Y-maze test with WT and KI/+ mice (8-week-old, male, $n = 13$ mice per group) 20 h after LPS (0.35 μg/kg body weight) i.p. injection for assessment of spatial working memory (KI/+ vs LPS-injected KI/+: †$p = 0.0391$; LPS-injected WT vs LPS-injected KI/+: ##$p = 0.0029$; one-way ANOVA). **f**, **g** T-maze test with WT and KI/+ mice (8-week-old, male, $n = 11$ mice per group) 20 h after LPS (0.35 μg/kg body weight) i.p. injection for assessment of spatial learning and memory (KI/+ vs LPS-injected KI/+: †††$p < 0.0001$; LPS-injected WT vs LPS-injected KI/+: ###$p < 0.0001$; one-way ANOVA). Tracking images indicate mouse head (blue line), center (red line), and tail (purple). The bounds of the box represent 25th to 75th percentiles ranges. The center lines with the box represent the mean value and whiskers represent the minimum to the maximum range (**c–g**). Data are mean ± SEM. †$p < 0.05$, and †††$p < 0.001$ vs untreated KI/+ control. ##$p < 0.01$, and ###$p < 0.001$ for indicated comparisons. Source data are provided as a Source Data file.

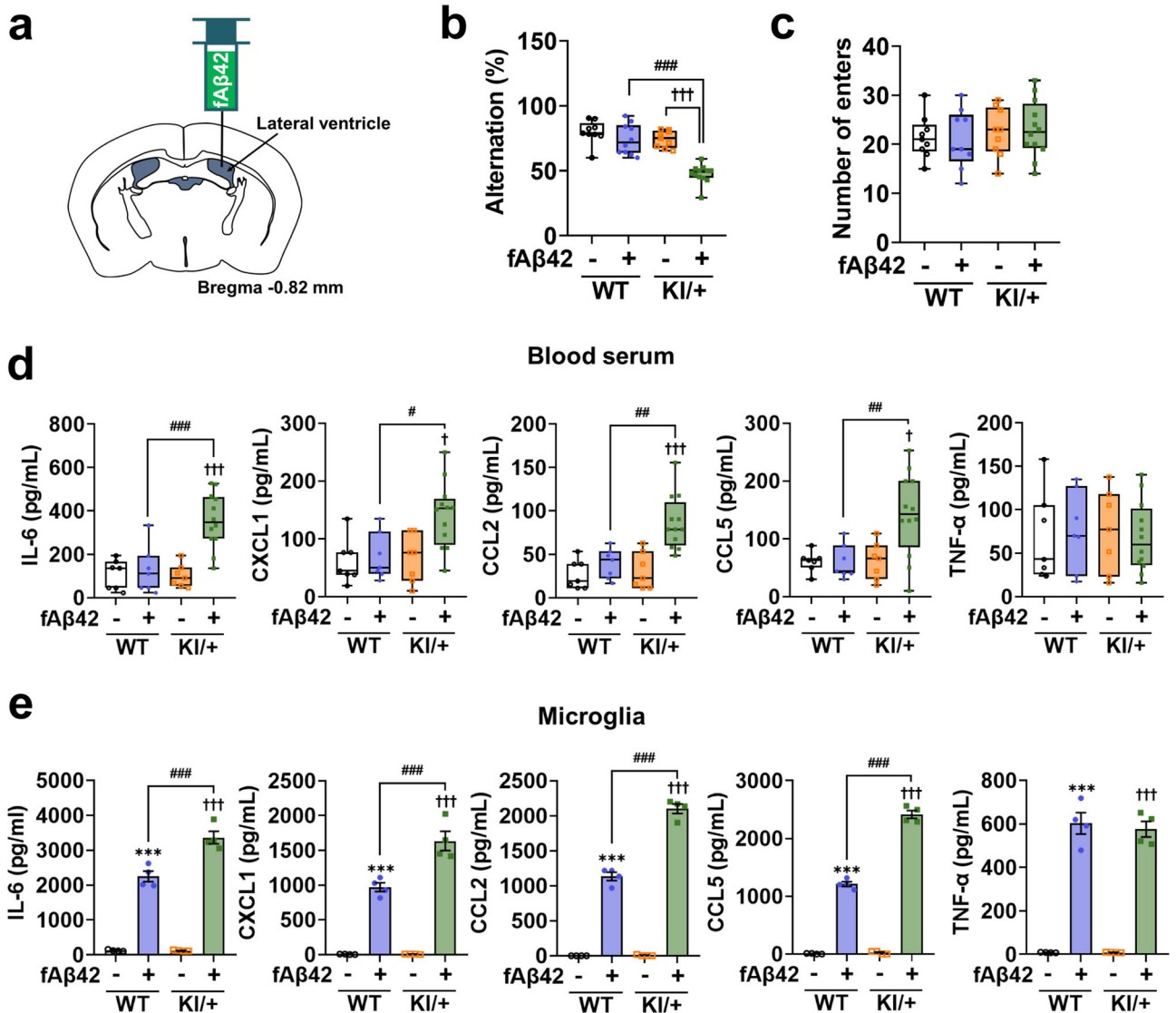

**Fig. 4 Fibrillar amyloid beta42 (fAβ42) increases secretion of clock gene-controlled cytokines and induces memory deficits in *Psen2*^N141I/+ mice.**
**a** Coordination of the stereotaxic surgery. **b**, **c** Y-maze test with WT and KI/+ mice (8-week-old, male, $n = 12$ mice for fAβ42-injected KI/+; $n = 7$ mice for other groups) 7 days after fAβ42 (1.25 nM) i.c.v. injection for assessment of spatial working memory (KI/+ vs fAβ42-injected KI/+: †††$p < 0.0001$; fAβ42-injected WT vs fAβ42-injected KI/+: ###$p < 0.0001$; one-way ANOVA). **d** ELISA of clock gene-controlled cytokines (IL-6, CXCL1, CCL2, and CCL5) and TNF-α in blood serum from WT and KI/+ mice (8-week-old, male, $n = 12$ mice for fAβ42-injected KI/+; $n = 7$ mice for other groups) 7 days after fAβ42 (1.25 nM) i.c.v. injection (KI/+ vs fAβ42-injected KI/+: †††$p < 0.0001$, †$p = 0.0109$, †††$p = 0.0002$, and †$p = 0.0103$; fAβ42-injected WT vs fAβ42-injected KI/+: ###$p = 0.0003$, #$p = 0.016$, ##$p = 0.0014$, and ##$p = 0.0087$; one-way ANOVA). The bounds of the box represent 25th to 75th percentiles ranges. The center lines with the box represent the mean value and whiskers represent the minimum to the maximum range (**b**–**d**). **e** ELISA of clock gene-controlled cytokines (IL-6, CXCL1, CCL2, and CCL5) and TNF-α in the culture media of WT and KI/+ microglia after fAβ42 (4 μM) treatment for 24 h ($n = 4$) (untreated WT vs fAβ42-treated WT: ***$p < 0.0001$; untreated KI/+ vs fAβ42-treated KI/+: †††$p < 0.0001$; fAβ42-treated WT vs fAβ42-treated KI/+: ###$p = 0.0001$, ###$p = 0.0003$, ###$p < 0.0001$, and ###$p < 0.0001$; one-way ANOVA). Data are mean ± SEM. ***$p < 0.001$ vs untreated WT control. †$p < 0.05$, and †††$p < 0.001$ vs untreated KI/+ control. #$p < 0.05$, ##$p < 0.01$, and ###$p < 0.001$ for the indicated comparisons. Source data are provided as a Source Data file.

report[39], the chromatin immunoprecipitation (ChIP) assay showed that REV-ERBα bound directly to the proximal RORE (AGGTCA) promoter site of the *Il-6* but not *Tnf* gene (Fig. 6b). Promoters of *Arntl* and the non-specific TATA box-binding protein (*Tbp*) gene were used as a positive and negative control of promoter binding by REV-ERBα, respectively[41]. The REV-ERBα binding level was significantly lower in KI/+ microglia than in WT (Fig. 6b), suggesting that downregulation of REV-ERBα may underlie the marked increase in IL-6 response to LPS. To confirm this notion, we knocked down the *Nr1d1* gene in WT microglia by transduction with lentivirus expressing *Nr1d1*-targeting

shRNA (Fig. 6c). Similar to the *Psen2* N141I mutation, suppression of *Nr1d1* expression in WT microglia potentiated the inflammatory response, as shown by an increase in the secretion (Fig. 6d) and mRNA expression (Fig. 6e) of IL-6 in response to LPS. In addition, the mRNA (Supplementary Fig. 5a) and secretion levels (Supplementary Fig. 5b) of other clock gene-controlled cytokines (CXCL1, CCL2, and CCL5) were increased with the suppression of *Nr1d1*. Thus, the knockdown of REV-ERBα was sufficient to induce a selective increase in the production of clock gene-controlled cytokines. We next over-expressed REV-ERBα in KI/+ microglia using lentivirus (Fig. 6f).

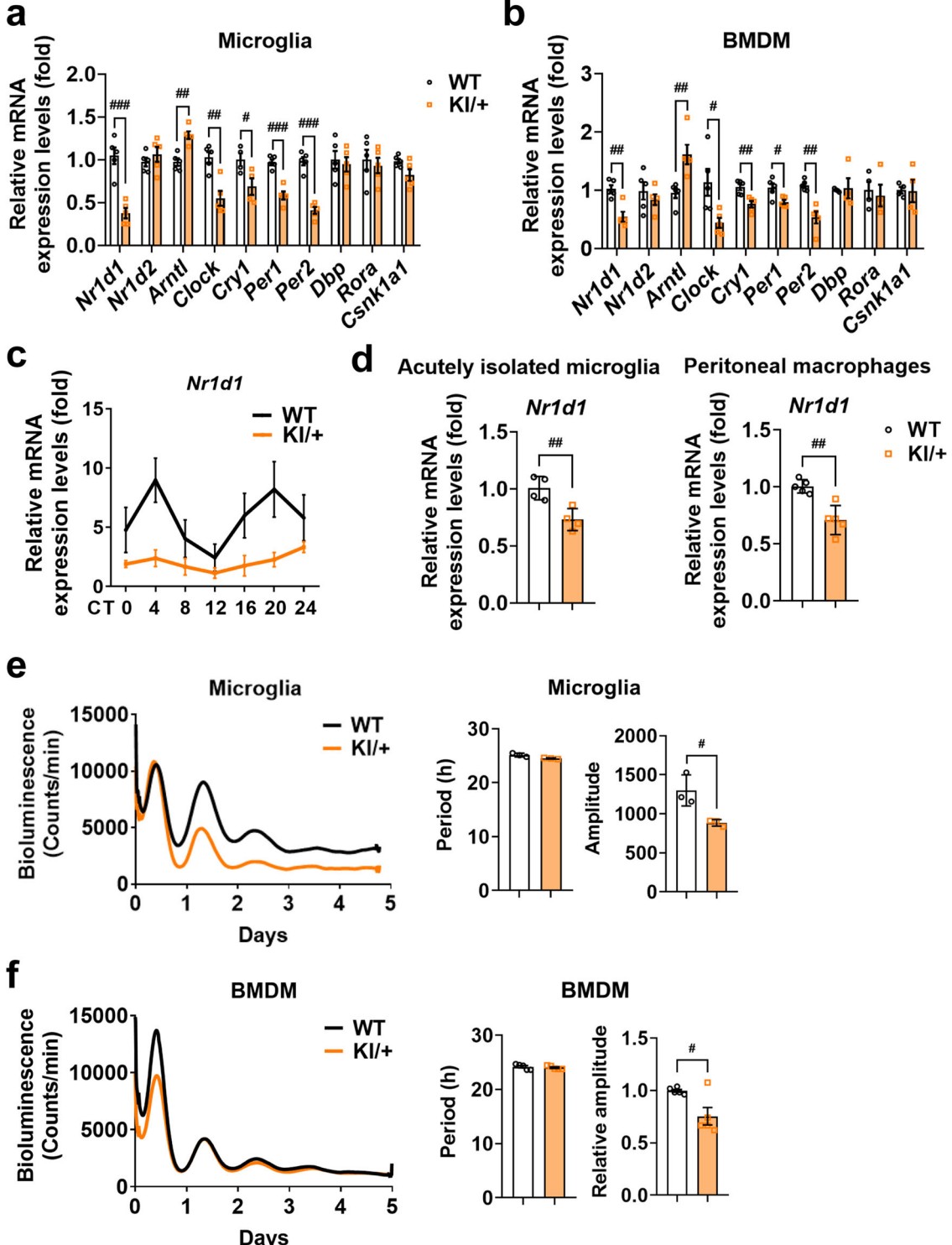

As expected, the expression of human *NR1D1* in KI/+ microglia was sufficient to revert the response of IL-6 to LPS to the WT levels (Fig. 6g, h). In addition, the mRNA (Supplementary Fig. 5c) and secretion levels (Supplementary Fig. 5d) of other clock-controlled cytokines were reduced with overexpression of *Nr1d1*. In contrast, the secretion and transcript level of TNF-α were not impacted by modulating REV-ERBα expression in microglia (Fig. 6d, e, g, h). These data indicate that the *Psen2* N141I mutation selectively elevates clock-controlled cytokines production in response to LPS through downregulation of REV-ERBα in microglia.

**Nr1d1 promoter is hypermethylated in KI/+ innate immune cells.** Recently, the role of epigenetic mechanisms in the pathogenesis of AD has been gaining attention[42]. Notably, the methylation of DNA, which occurs mostly on CpG dinucleotides (CpG islands) and results in transcriptional repression, is altered in AD brains[43]. To determine whether *Psen2* N141I mutation represses *Nr1d1* expression via DNA hypermethylation, we performed whole-genome DNA methylation sequencing of WT and KI/+ microglia. Next, we searched CpG islands in the *Nr1d1* promoter using the prediction algorithm at www.urogene.org/methprimer and estimated methylation levels in the predicted

**Fig. 5 Innate immune cells derived from *Psen2*[N141I/+] mice are rhythmic despite suppressed expression of clock genes. a, b** Comparison of relative mRNA expression levels of clock genes between WT and KI/+ primary microglia (**a**; $n = 4$ for *Cry1*; $n = 5$ for other genes) (WT vs KI/+: [###]$p = 0.0003$, [##]$p = 0.0011$, [##]$p = 0.0021$, [#]$p = 0.042$, [###]$p = 0.0001$, and [###]$p < 0.0001$) and BMDM cultures (**b**) ($n = 4$ for *Dbp*, *Rora* and *Csnk1a1*; $n = 5$ for other genes) (WT vs KI/+: [##]$p = 0.0022$, [##]$p = 0.008$, [#]$p = 0.0221$, [##]$p = 0.0049$, [#]$p = 0.0181$, and [##]$p = 0.001$; unpaired *t*-test). **c** Analyses of transcript abundance of *Nr1d1* at different circadian times (CT). Microglia were synchronized with dexamethasone (DEX, 100 nM) for 2 h and harvested every 4 h for RNA extraction and qPCR ($n = 4$). **d** Comparison of relative mRNA expression levels of *Nr1d1* in acutely isolated microglia ($n = 4$ mice) ([##]$p = 0.0078$) and peritoneal macrophages ($n = 5$ mice) ([##]$p = 0.0015$) between WT and KI/+ mice (8-week-old, male). mRNA levels were normalized to 18s rRNA. **e, f** Representative bioluminescence recording of rhythmic *Per2::Luc* expression in primary microglia (**e**) and BMDM (**f**) from *Per2::Luc;Psen2*[+/+] or *Per2::Luc;Psen2*[N141I/+] mice. Cells were cultured and exposed to DEX (100 nM) for 2 h, followed by measurement of luciferase bioluminescence (each measurement for 1 min with 10-min interval) for 5 days. Circadian period and amplitude were analyzed in microglia (**e**; $n = 3$) ([#]$p = 0.0245$) and BMDM (**f**; $n = 5$) ([#]$p = 0.02$). Unpaired *t*-test for analysis. Data are mean ± SEM. [#]$p < 0.05$, [##]$p < 0.01$, and [###]$p < 0.001$ for the indicated comparisons. Source data are provided as a Source Data file.

CpG islands through differential cytosine methylation analysis. As a result, the overall methylation ratio of *Nr1d1* ~2 kbp upstream of the transcription start site was significantly higher in KI/+ microglia than in WT (Fig. 7a). On the contrary, the same differential cytosine methylation analysis of the promoters of other clock genes revealed similar levels of DNA methylation (Fig. 7b).

To examine whether promoter methylation is associated with REV-ERBα expression, we demethylated DNA in KI/+ microglia using 5-azacytidine (5-Aza) and determined *Nr1d1* transcript level by qRT-PCR. Treatment of KI/+ microglia with 5-Aza for 24 h rescued *Nr1d1* mRNA (Fig. 7c) and REV-ERBα protein levels (Fig. 7d), and subsequently decreased IL-6 expression (Fig. 7e) and secretion (Fig. 7f) to the levels similar to those in WT, without affecting TNF-α expression (Fig. 7e) or secretion (Fig. 7f). The same treatment with 5-Aza also restored the *Nr1d1* transcript (Fig. 7c) and REV-ERBα protein levels (Supplementary Fig. 6a), and reverted elevated expression and secretion profiles of IL-6 without affecting TNF-α in KI/+ BMDM (Supplementary Fig. 6b, c). 5-Aza treatment had no effects in WT cells (Fig. 7c–f and Supplementary Fig. 6a–c). Taken together, these data suggest that DNA hypermethylation of the promoter of the *Nr1d1* gene is critical for reducing its expression in KI/+ innate immune cells.

**The antipsychotic chlorpromazine restores REV-ERBα expression, and prevents hyperactive immune response and cognitive decline in *Psen2*[N141I/+] mice.** Aiming at restoring the REV-ERBα level and thereby preventing the hyperactive immune response, we screened a chemical library containing 2150 compounds, which consists of clinical phase I-III compounds and FDA-approved, commercially available drugs. We chose compounds that preferentially reduced IL-6 secretion over TNF-α, and selected compounds for further testing after eliminating compounds showing cytotoxicity and already known to have anti-inflammatory or anti-AD effects. Among selected compounds, we narrowed the list to the compounds that are predicted to cross the blood–brain barrier (Supplementary Fig. 7a).

Among the drugs filtered, the antipsychotic chlorpromazine (CPZ, 0.5 μM for microglia and 1 μM for BMDM) efficiently reduced IL-6 secretion and mRNA level (Fig. 8a and Supplementary Fig. 7b) without affecting those of TNF-α (Supplementary Fig. 7c, d) in KI/+ microglia and BMDM. Also, CPZ reversed LPS-induced increases in the mRNA levels of other clock gene-controlled cytokines in KI/+ microglia and BMDM (Supplementary Fig. 7e–g). At these concentrations, CPZ was not toxic to the cells (cell death of 2.76% ± 1.78% in WT and 2.39% ± 0.42% in KI/+ microglia; 2.4% ± 0.69% in WT and 2.26% ± 0.19% in KI/+ BMDM). Furthermore, CPZ treatment of KI/+ microglia rescued *Nr1d1* mRNA (Fig. 8c) and protein levels (Fig. 8d). CPZ treatment of KI/+ BMDM also restored *Nr1d1* mRNA (Supplementary Fig. 7h) and protein levels (Supplementary Fig. 7i).

Moreover, CPZ treatment in KI/+ microglia rescued fAβ42-induced hyperimmunity without affecting TNF-α levels (Fig. 8e). Of note, CPZ treatment did not alter mRNA levels of other clock genes in KI/+ microglia and BMDM as well as WT cells (Supplementary Fig. 8a, b).

To examine whether CPZ can attenuate the overproduction of clock gene-controlled cytokines in vivo, we injected CPZ into WT and KI/+ mice 4 h before LPS injection (Fig. 8f). Administration of CPZ decreased the blood levels of clock gene-controlled cytokines in KI/+ mice without affecting TNF-α (Fig. 8g). To examine whether CPZ can rescue LPS-induced memory deficits in KI/+ mice, we conducted Y-maze test (Fig. 8h, i). CPZ recovered the arm alternation in LPS-injected KI/+ mice (Fig. 8h). The total number of arm entries was the same among all groups, indicating that CPZ-injected mice had normal locomotor activity (Fig. 8i).

In addition, CPZ injection in KI/+ mice recovered microglial morphology (Fig. 9a) and rescued the total dendrite length, the number of dendrite terminal points and dendrite intersections at the basal state and prevented LPS-induced morphological changes (Fig. 9b, c). CPZ at the current low dose did not alter microglial morphology in WT mice (Supplementary Fig. 8c, d). Taken together, these data show that CPZ prevents LPS-induced hyperactive inflammation and memory deficits in *Psen2*[N141I/+] mice, revealing a previously unexplored action of this antipsychotic drug CPZ.

**CPZ rescues REV-ERBα expression by rectifying DNA hypermethylation through the inhibition of PI3K/AKT1 pathway in KI/+ microglia.** To examine whether CPZ restores *Nr1d1* expression via demethylation of its promoter, we performed methylation-specific PCR on the *Nr1d1* promoter. We designed methylation primers based on the regions showing higher methylation ratios in KI/+ microglia (Fig. 7a). Hypermethylation was observed in genomic DNA extracted from KI/+ microglia (Fig. 10a) and CPZ treatment effectively demethylated *Nr1d1* promoter in KI/+ microglia (Fig. 10a). 5-Aza was used as a positive control for demethylation (Fig. 10a). DNA methylation is regulated by the enzymes called DNA methyltransferases (DNMT) and ten-eleven translocation (TET). DNMTs convert unmodified cytosine to 5-methylcytosine (5mC), while TET enzymes convert back the oxidized 5mC into 5-hydroxymethylcytosine and thereby catalyze demethylation[44]. Thus, we examined the mRNA expression levels of DNMTs and TETs and observed higher levels of *Dnmt3a* and *Dnmt3b* while *Tet1* was lower in KI/+ microglia than WT cells (Fig. 10b). Similarly, mRNA expression levels of *Dnmt3a* and *Dnmt3b* were upregulated but *Tet1* was downregulated in acutely isolated microglia (Supplementary Fig. 9a) and peritoneal macrophages (Supplementary Fig. 9b). Importantly, CPZ reverted a significantly high expression level of *Dnmt3a* to that of WT (Fig. 10c) and downregulated its protein amount (Fig. 10d).

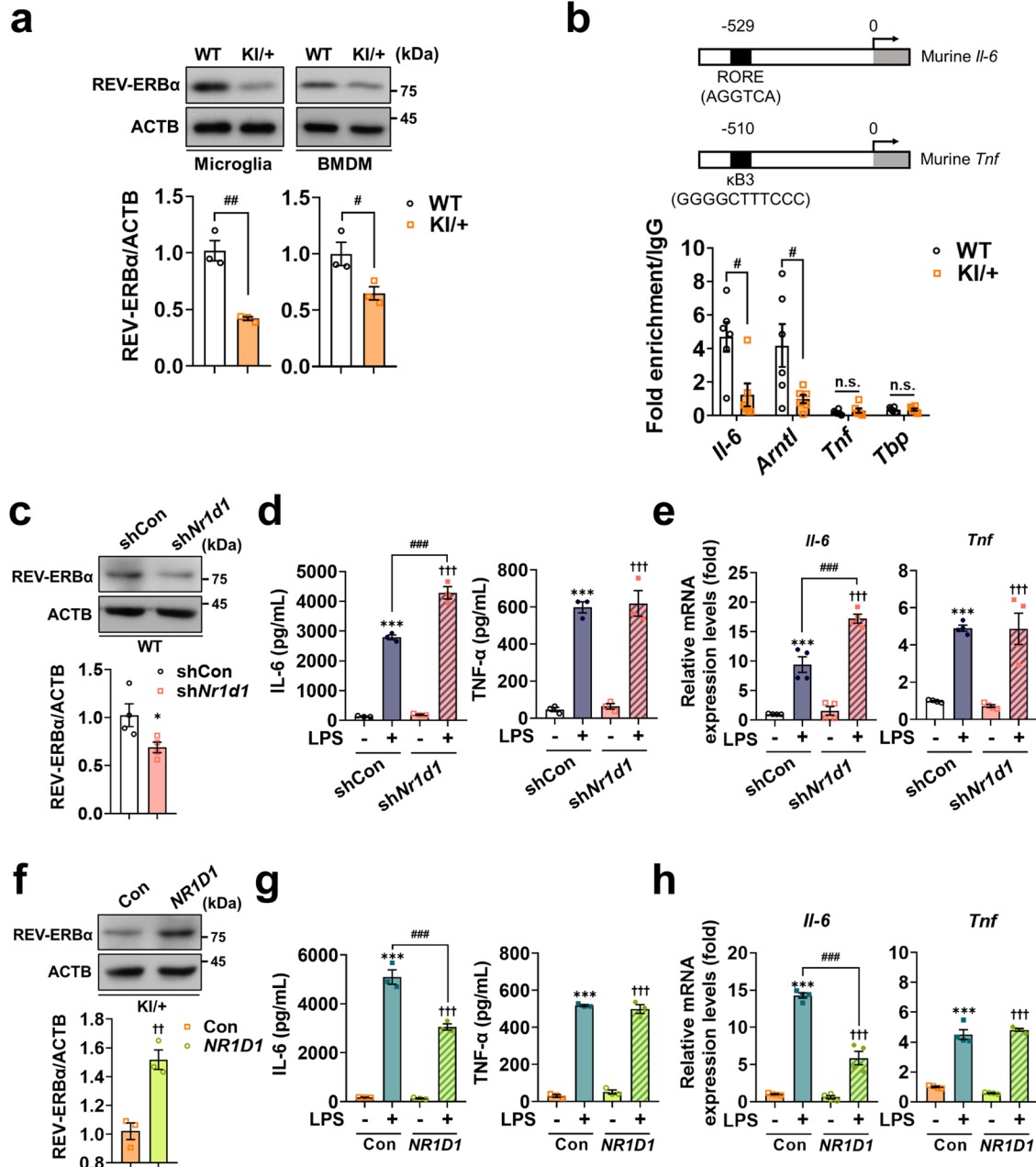

**Fig. 6 *Psen2* N141I mutation selectively elevates production of clock-controlled cytokines through the downregulation of REV-ERBα in innate immune cells. a** Analyses of REV-ERBα protein levels by western blotting in microglia ($n = 3$) ($^{##}p = 0.0028$) and BMDM ($n = 3$) ($^{#}p = 0.042$). **b** Primer target regions. Average fold enrichment of immunoprecipitated DNA fragments in microglia was analyzed by qRT-PCR in ChIP assay. Average fold enrichment was normalized to IgG of each gene ($n = 6$) (WT vs KI/+: $^{#}p = 0.0106$ and $^{#}p = 0.0338$; n.s., not significant; unpaired $t$-test). **c** Relative REV-ERBα expression levels in microglia transduced with lentivirus expressing shCon or sh*Nr1d1* ($n = 4$) ($^{*}p = 0.0423$; unpaired $t$-test). **d**, **e** Secretion (**d**; $n = 3$) and mRNA levels (**e**; $n = 4$) of IL-6 and TNF-α in WT microglia transduced with sh*Nr1d1* after LPS (1 μg/mL) treatment for 12 h (untreated WT vs LPS-treated WT: $^{***}p < 0.0001$, $^{***}p < 0.0001$, $^{***}p < 0.0001$, and $^{***}p = 0.0002$; untreated KI/+ vs LPS-treated KI/+: $^{†††}p < 0.0001$; LPS-treated WT vs LPS-treated KI/+: $^{###}p < 0.0001$ and $^{###}p = 0.0001$; one-way ANOVA). **f** Relative REV-ERBα expression levels in KI/+ microglia expressing human *NR1D1* ($n = 3$) ($^{††}p = 0.005$; unpaired $t$-test). **g**, **h** Secretion (**g**; $n = 3$) and mRNA levels (**h**; $n = 4$) of IL-6 and TNF-α in *NR1D1*-overexpressing KI/+ microglia after LPS (1 μg/mL) treatment for 12 h (untreated WT vs LPS-treated WT: $^{***}p < 0.0001$; untreated KI/+ vs LPS-treated KI/+: $^{†††}p < 0.0001$; LPS-treated WT vs LPS-treated KI/+: $^{###}p < 0.0001$; one-way ANOVA). Signals on western blots were normalized to those of ACTB. mRNA levels were normalized to *Actb*. Data are mean ± SEM. $^{*}p < 0.05$, and $^{***}p < 0.001$ vs untreated WT control. $^{††}p < 0.01$, and $^{†††}p < 0.001$ vs untreated KI/+ control. $^{#}p < 0.05$, $^{##}p < 0.01$, and $^{###}p < 0.001$ for the indicated comparisons. Source data are provided as a Source Data file.

To identify the signaling pathway responsible for the DNA hypermethylation, we focused on MAPK1/3, GSK3β, and PI3K/ ATK1 since they are known to regulate DNMT3A expression[45–48]. All these kinases were highly phosphorylated in KI/+ microglia

(Fig. 10e), but only phosphorylation of AKT1 was reduced by CPZ (Fig. 10f).

Consistently, pharmacological inhibition of PI3K downregu- lated DNMT3A and rescued *Nr1d1* level (Fig. 10g, h). However,

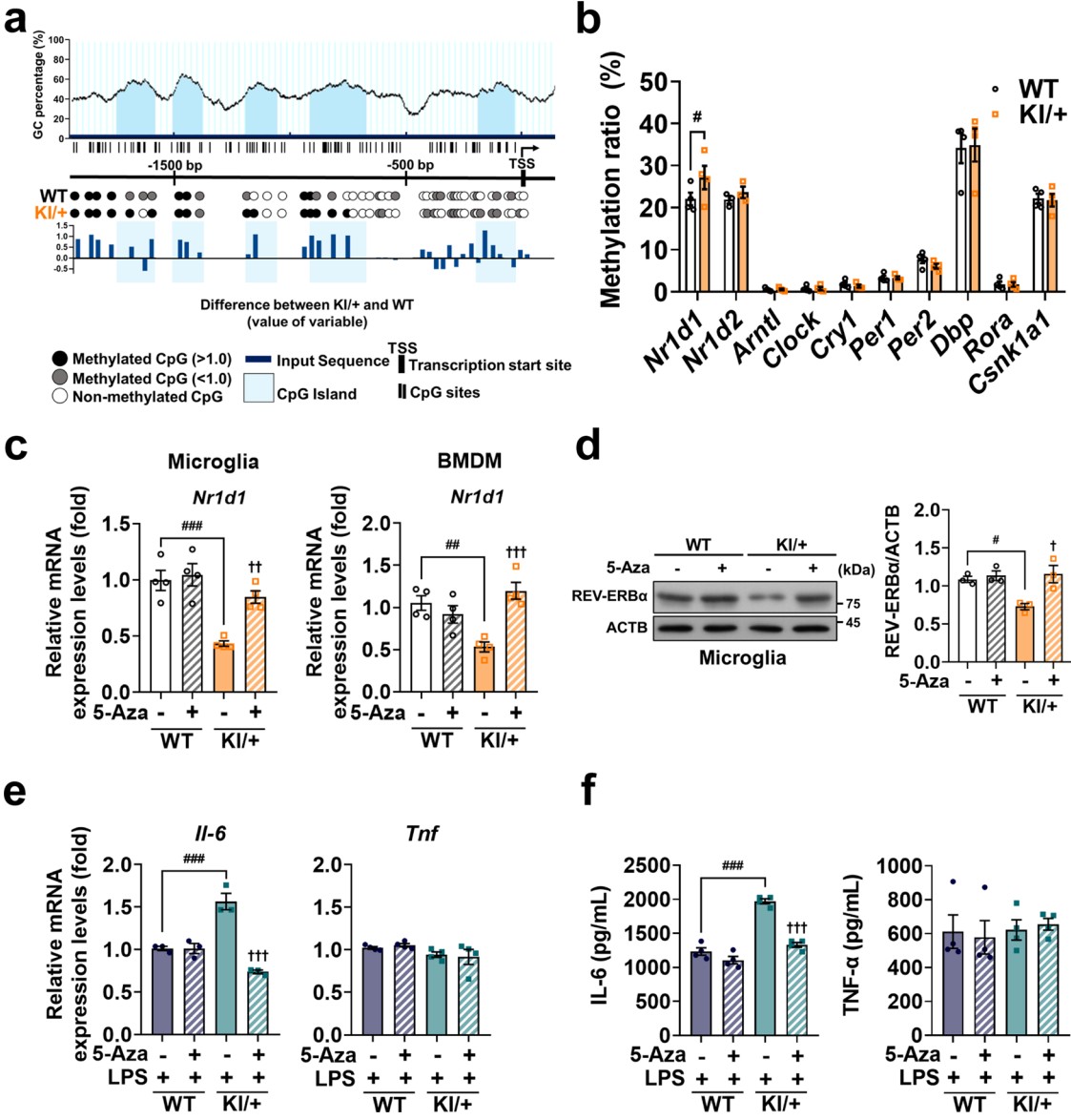

**Fig. 7 DNA hypermethylation underlies the repression of REV-ERBα in *Psen2*$^{N141I/+}$ immune cells. a** Methylation analysis of CpG islands in the *Nr1d1* promoter. Difference in the value of the sum of methylation rate (calculated as methylation depth / (methylation depth + non-methylation depth)) between KI/+ and WT at each methylated CpG site is shown as a graph with a blue bar for each position. **b** Comparison of overall DNA methylation ratios in the 2-kbp promoter region of clock genes between KI/+ and WT microglia ($n = 4$) (#$p = 0.0383$; unpaired *t*-test). **c, d** Pharmacological modulation with 5-Azacytidine (5-Aza, 10 μM) treatment for 24 h restores *Nr1d1* transcript in KI/+ microglia and BMDM (**c**; $n = 4$, respectively) and protein levels in KI/+ microglia (**d**; $n = 3$) (untreated WT vs untreated KI/+: ###$p = 0.0008$, ##$p = 0.0062$, and #$p = 0.03$; untreated KI/+ vs 5-Aza-treated KI/+: ††$p = 0.0084$, †††$p = 0.0009$, and †$p = 0.0118$; one-way ANOVA). Signals on western blots were normalized to those of ACTB. **e, f** 5-Aza-induced demethylation reduces the mRNA level (**e**; $n = 3$ for *Il-6*; $n = 4$ for *Tnf*) and secretion (**f**; $n = 4$) of IL-6 but not TNF-α in KI/+ microglia (LPS-treated WT vs LPS-treated KI/+: ###$p = 0.0008$ and ###$p < 0.0001$; LPS-treated KI/+ vs LPS and 5-Aza-treated KI/+: †††$p < 0.0001$; one-way ANOVA). Cells were pre-treated with 5-Aza for 12 h, and mRNA levels and secretion were analyzed after co-treatment with LPS for 12 h. mRNA levels were normalized to *Actb*. Data are mean ± SEM. †$p < 0.05$, ††$p < 0.01$, and †††$p < 0.001$ vs untreated KI/+ control. #$p < 0.05$, ##$p < 0.01$, and ###$p < 0.001$ for the indicated comparisons. Source data are provided as a Source Data file.

inhibition of MAPK1/3 by PD98059, as shown by the decreased phosphorylation of MAPK1/3 at T202/Y204 (Supplementary Fig.10a), did not affect AKT1 phosphorylation and failed to rescue DNMT3A and *Nr1d1* levels in KI/+ microglia (Supplementary Fig. 10a, b). An increase in the inhibitory phosphorylation of S9 indicates decreased GK3β activity in KI/+ microglia (Fig. 10e). There are only a few GSK3β activators reported. Differentiation-inducing factor-3 (DIF-3) was shown to activate GSK3β independently of AKT1 or MAPK cascade[49]. So, we used DIF-3 to examine whether GSK3β activation can rescue *Nr1d1*

suppression in KI/+ microglia. Low dose (5 μM) of DIF-3 did not activate GSK3β (Supplementary Fig. 10d). DIF-3 at the concentration of 7.5 μM was able to activate GSK3β, as shown by the dephosphorylation of S9, but did not affect AKT1 phosphorylation and failed to rescue DNMT3A and *Nr1d1* levels in KI/+ microglia (Supplementary Fig. 10d, e). We treated primary microglia with DIF-3 at this concentration for 3 h due to the cytotoxicity after longer treatment, and DIF-3 at higher doses than 7.5 μM was highly toxic to microglia. PD and DIF-3 decreased mRNA levels of *Il-6* and *Tnf* in both WT and KI/+ microglia in a

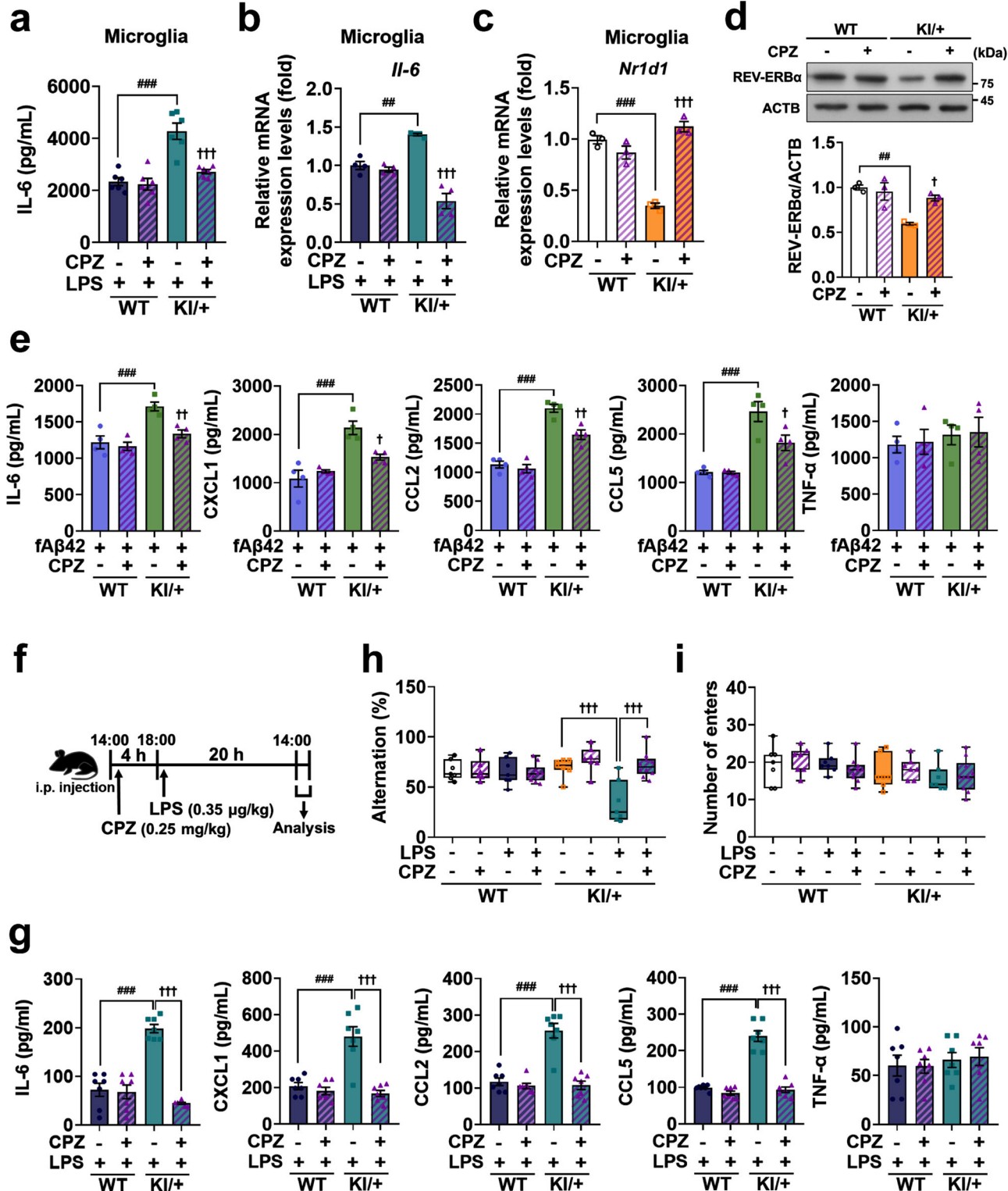

dose-dependent manner (Supplementary Fig. 10c, f). Therefore MAPK1/3 or GSK3β was not closely associated with PI3K/AKT1 overactivation under our experimental condition. These data indicate that the overactivation of PI3K/AKT1 pathway is the proximal event involved in *Nr1d1* suppression in KI/+ microglia. To sum up, CPZ restored REV-ERBα expression levels by correcting DNA hypermethylation through the inhibition of PI3K/AKT1 pathway in *Psen2* N141I KI/+ microglia.

## Discussion

Here, we show that *Psen2* N141I mutation downregulates REV-ERBα in innate immune cells and mutant mice fail to optimize the innate immune response to an otherwise innocuous immune and fAβ42 challenges, leading to selective overproduction of clock gene-controlled cytokines and cognitive decline (Fig. 10i). We revealed hypermethylation of the *Nr1d1* promoter through the PI3K/AKT1 pathway and upregulation of DNMTs as the

**Fig. 8 Chlorpromazine prevents hyperimmune response and cognitive decline in P*sen2*[N141I/+] mice. a, b** Chlorpromazine (CPZ, 0.5 μM) treatment reduces the secretion (**a**; $n = 6$) and mRNA level (**b**; $n = 4$) of IL-6 in KI/+ microglia-treated with LPS for 12 h (LPS-treated WT vs LPS-treated KI/+: [###]$p < 0.0001$ and [##]$p = 0.0019$; LPS-treated WT vs LPS and CPZ-treated KI/+: [†††]$p = 0.0002$ and [†††]$p < 0.0001$; one-way ANOVA). **c, d** CPZ treatment restores *Nr1d1* transcript (**c**; $n = 3$) and protein levels (**d**; $n = 3$) in KI/+ microglia-treated with LPS for 12 h (untreated WT vs untreated KI/+: [###]$p < 0.0001$ and [##]$p = 0.0029$; untreated WT vs CPZ-treated KI/+: [†††]$p < 0.0001$ and [†]$p = 0.0214$; one-way ANOVA). mRNA levels were normalized to *Actb*. **e** CPZ treatment reduces secretion of IL-6, CXCL1, CCL2, CCL5, and TNF-α in KI/+ microglia treated with fAβ42 (4 μM) for 24 h ($n = 4$) (fAβ42-treated WT vs fAβ42-treated KI/+: [###]$p = 0.0009$, [###]$p = 0.0002$, [###]$p < 0.0001$, and [###]$p = 0.0001$; fAβ42-treated KI/+ vs fAβ42 and CPZ-treated KI/+: [††]$p = 0.0077$, [†]$p = 0.0144$, [††]$p = 0.0032$, and [†]$p = 0.0217$; one-way ANOVA). CPZ was treated 30 min prior to LPS (**a–e**). **f** Schematic illustration of the experimental schedule. WT and KI/+ mice were analyzed 20 h after i.p. injection of LPS (0.35 μg/kg). CPZ (0.25 mg/kg) was administered i.p. 4 h prior to LPS injection. **g** ELISA of IL-6, CXCL1, CCL2, CCL5, and TNF-α in blood serum from WT and KI/+ mice (8-week-old, male, $n = 7$ mice per group) (LPS-injected WT vs LPS-injected KI/+: [###]$p < 0.0001$; LPS-injected KI/+ vs LPS and CPZ-injected KI/+: [†††]$p < 0.0001$; one-way ANOVA). **h, i** Y-maze test with WT and KI/+ mice (8-week-old, male, $n = 10$ mice for LPS and CPZ-injected group; $n = 7$ mice for other groups) with arm alternation (**h**) and number of arm entries (**i**) (KI/+ vs LPS-injected KI/+: [†††]$p < 0.0001$; LPS-injected KI/+ vs LPS and CPZ-injected KI/+: [†††]$p < 0.0001$; one-way ANOVA). The bounds of the box represent 25th to 75th percentiles ranges. The center lines with the box represent the mean value and whiskers represent the minimum to the maximum range. Data are mean ± SEM. [†]$p < 0.05$, [††]$p < 0.01$, and [†††]$p < 0.001$ vs untreated KI/+ control. [##]$p < 0.01$, and [###]$p < 0.001$ for the indicated comparisons. Source data are provided as a Source Data file.

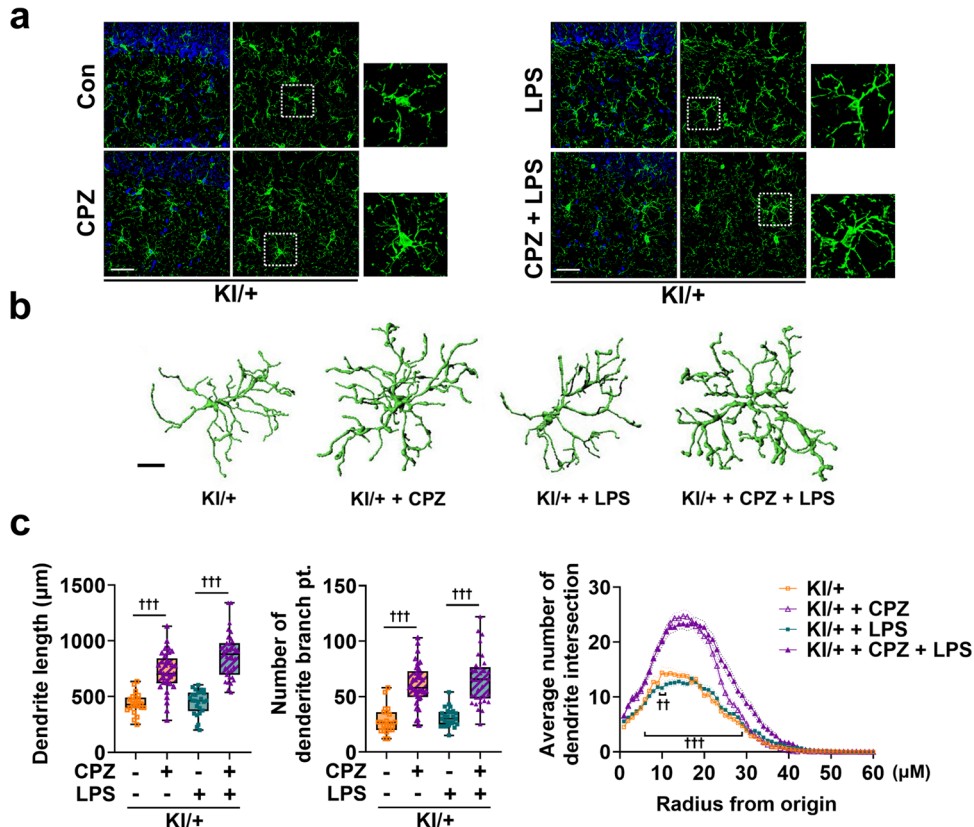

**Fig. 9 Chlorpromazine recovers microglial morphology. a** Representative immunofluorescent images by IBA-1 staining in the hippocampus of WT and KI/+ mice (8-week-old, male, $n = 4$ mice with similar results). Insets, 2.5-fold enlarged images. Scale bar, 25 μm. **b** Representative 3D filament tracking images of IBA-1 signals generated by IMARIS software. Scale bar, 10 μm. **c** Dendrite length, number of branching branch points, and Sholl radius analysis. Data extracted from FilamentTracker in IMARIS analysis ($n > 22$ cells per group from four mice) (KI/+ vs CPZ-injected KI/+: [†††]$p < 0.0001$; LPS-injected KI/+ vs LPS and CPZ-injected KI/+: [†††]$p < 0.0001$; one-way ANOVA; radius 10 μm, KI/+ vs LPS-injected KI/+: [††]$p = 0.0049$; radius 6–29 μm, LPS-injected KI/+ vs LPS and CPZ-injected KI/+: [†††]$p < 0.0001$; two-way ANOVA). The bounds of the box represent 25th to 75th percentiles ranges. The center lines with the box represent the mean value and whiskers represent the minimum to the maximum range. Data are mean ± SEM. [††]$p < 0.01$, and [†††]$p < 0.001$ vs untreated KI/+ control. Source data are provided as a Source Data file.

underlying epigenetic repressive mechanism. Therefore, our findings highlight a mechanism linking the *Psen2* N141I FAD mutation to altered expression of REV-ERBα, which renders mutant mice vulnerable to AD immunopathology.

In addition to *Nr1d1*, other clock genes including *Per1*, *Per2*, *Cry*, and *Clock* were also markedly downregulated in KI/+ microglia and BMDM. However, overactive response of IL-6 and other clock gene-controlled cytokines is not readily explained by their downregulation. CLOCK forms a complex with the NF-κB subunit p65 and increases the transcriptional activity of NF-κB[50]. Therefore, Clock-deficient BMDM is less responsive to LPS[51]. PER2 can also promote inflammation, and *Per2*[−/−] mice are protected from LPS-induced endotoxic shock and show decreased serum levels of several cytokines, but the IL-6 level remains normal[52]. *Cry1*[−/−]*Cry2*[−/−] BMDM show increased inflammatory features, but without selectivity for IL-6 or clock gene-controlled

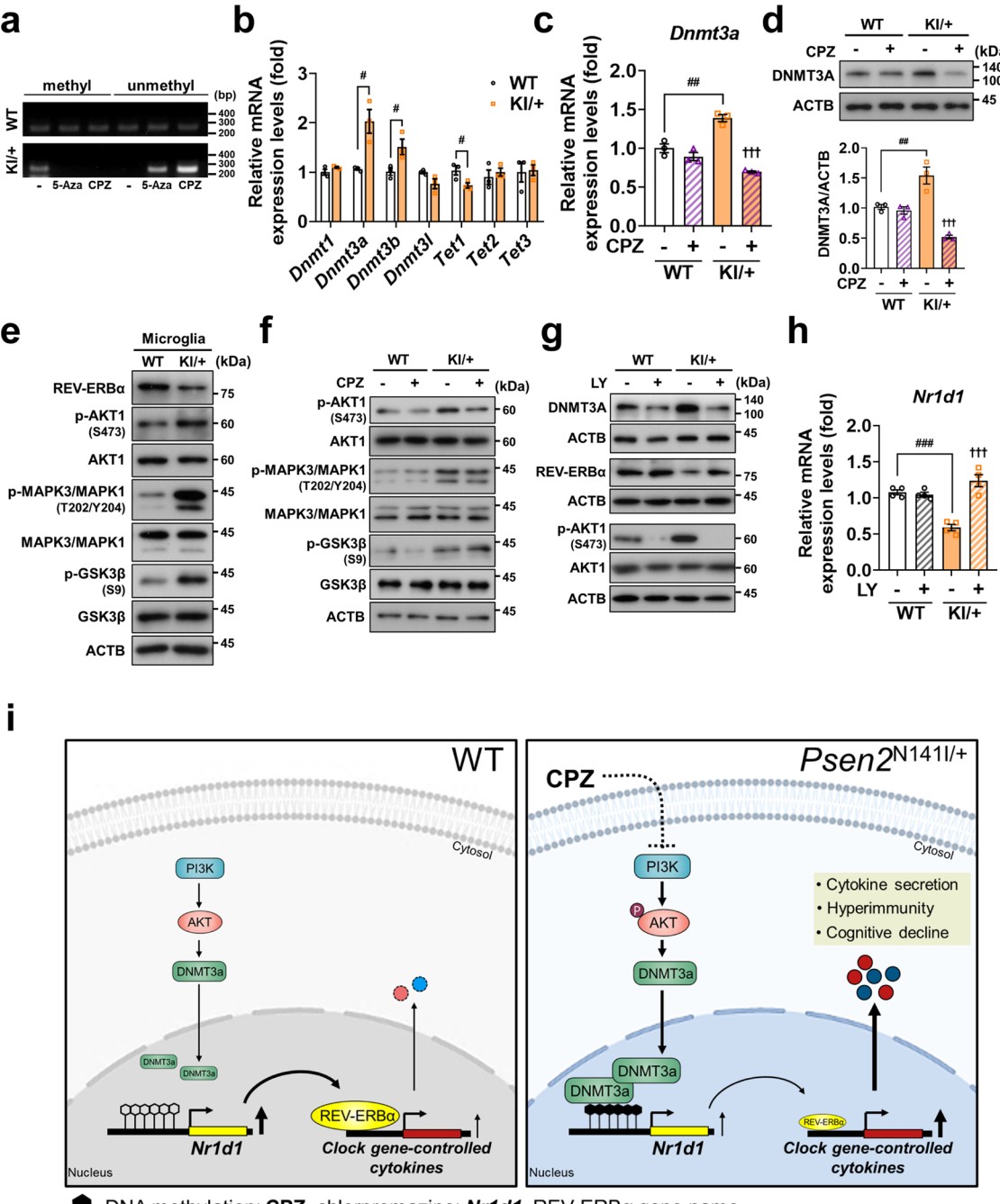

**Fig. 10 Chlorpromazine recovers REV-ERBα level by rectifying hypermethylation of the *Nr1d1* promoter through the inhibition of PI3K/AKT1 pathway in *Psen2* N141I KI/+ microglia. a** Methylation-specific PCR analysis on *Nr1d1* promoter of KI/+ microglia after CPZ (0.5 μM) treatment for 12 h. 5-Aza (10 μM) as a positive control was treated for 24 h. Genomic DNA extracted from microglia was conversed using bisulfite administration. The agarose gel image is representative of three independent experiments with similar results. **b** Comparison of relative mRNA expression levels of DNA (de)methylases between WT and KI/+ primary microglia (n = 3) (WT vs KI/+: #p = 0.0161, #p = 0.0447, and #p = 0.0459; unpaired t-test). **c, d** Relative *Dnmt3a* mRNA (**c**; n = 3) and protein levels (**d**; n = 3) after CPZ treatment for 12 h in WT and KI/+ microglia (untreated WT vs untreated KI/+: ##p = 0.0013 and ##p = 0.0083; untreated KI/+ vs CPZ-treated KI/+: †††p < 0.0001; one-way ANOVA). **e** Western blotting analyses of the phosphorylation levels of AKT1 at S473, MAPK1/3 at T202/Y204, and GSK3β at S9 in WT and KI/+ microglia. **f** CPZ inhibits PI3K/AKT pathway, but not MAPK1/3 or GSK3β. Cells were analyzed after CPZ (0.5 μM) treatment for 12 h. **g** LY (LY294002, 20 μM) inhibits PI3K/AKT pathway and downregulates DNMT3A leading to the restoration of REV-ERBα levels. The blots shown are representative of three experiments with similar results. **h** Relative *Nr1d1* mRNA expression levels after LY treatment for 12 h in WT and KI/+ microglia (n = 4). mRNA levels were normalized to 18s rRNA (**b**, **c**, **h**). **i** Schematic diagram suggesting hyperactive immune responses caused by the *Psen2* N141I mutation. The mutation activates PI3K/AKT pathway and upregulates DNMTs, which induces the epigenetic suppression of REV-ERBα and hyperactive immune response in innate immune cells. An antipsychotic, CPZ, rescues REV-ERBα expression by DNA demethylation through the inhibition of PI3K/AKT1 pathway, and prevents hyperimmunity and cognitive decline in *Psen2*ᴺ¹⁴¹ᴵ/⁺ mice. Data are mean ± SEM. †††p < 0.001 vs untreated KI/+ control. #p < 0.05, ##p < 0.01, and ###p < 0.001 for the indicated comparisons. Source data are provided as a Source Data file.

cytokines, and the levels of both IL-6 and TNF-α are increased by the absence of CRY[53]. Therefore, the selectively enhanced production of clock gene-controlled cytokines in $Psen2^{N141I/+}$ mice is most likely due to the repression of REV-ERBα. In addition, DNA methylation sequencing and differential cytosine methylation analysis in the promoter regions of clock genes revealed hypermethylation only in the promoter of $Nr1d1$ gene in KI/+ microglia. As such, CPZ recovered expression of only $Nr1d1$ among the clock genes, further lending support to our conclusion.

The mediation of circadian regulation by REV-ERBα in innate immune cells has been associated with inflammation in the periphery[32]. However, the role of REV-ERBα in central immunity has not been well known. Recently, the link between REV-ERBα and microglia function has been studied using gene knockout approaches[40]. Deletion of REV-ERBα activates NF-κB signaling, and multiple proinflammatory genes, including TNF-α and circadian-regulated cytokine genes, are upregulated, which is different from the selective elevation of expression of only circadian gene-controlled cytokines observed in this study. Also, $Psen2^{N141I/+}$ mice showed intact memory function and normal baseline behavior patterns (Figs. 3, 4, and 8 and Supplementary Fig. 3), which were different from the mildly disturbed circadian locomotor behaviors, neurodegeneration, or mood dysregulation of REV-ERBα KO mice[17,54]. Therefore, the repression of REV-ERBα due to the $Psen2$ N141I mutation did not phenocopy REV-ERBα KO mice, suggesting that the residual level of REV-ERBα is enough to maintain normal central clock and neural function. There has been great interest in understanding how circadian proteins go awry during the progress of neurodegeneration and the cause of their dysfunction. Our study uncovers the effects of the $Psen2$ N141I FAD mutation on the expression of REV-ERBα through unbalancing of DNA methylation/demethylation toward hypermethylation, and reveals a functionally important pathogenic pathway leading to hyperreactive innate immunity in AD.

CPZ was introduced into psychiatry for the treatment of schizophrenia and other psychoses in the 1950s[55]. Its psychotropic effects were mainly attributed to the blockade of dopamine receptors in neurons[56], and partially to the inhibition of neuroinflammation[57]. However, the action of CPZ on innate immune cells is largely unknown and the rescue of REV-ERBα expression reveals a previously unrecognized effect of CPZ.

It is known that antipsychotic drugs such as CPZ, when taken for a long time at high doses, increase mortality in elderly patients with dementia-related psychotic symptoms[58]. CPZ has been used in the concentration ranges of 1~25 mg/kg in previous animal studies[59–61]. However, in our study, we used CPZ at the dose of 0.35 µg/kg body weight, much lower than the reported doses. Therefore, further studies would be desirable to optimize the CPZ regimen for the prevention of immunopathology with reduced risk of toxicity.

$Psen2$ N141I mutation can alter various pathways, including autophagy[62], intracellular $Ca^{2+}$ level[63], and oxidative stress[64], all of which may lead to overactivation of PI3K/AKT1 pathway. However, the effects of $Psen2$ N141I mutation on these pathways in immunocompetent cells remain poorly understood. Interestingly, CPZ was identified as an autophagy inducer through a cell-based high-throughput image screening[65]. CPZ can negatively regulate store-operated $Ca^{2+}$ entry and inhibit proton channel[66,67]. These effects of CPZ may contribute to its immunomodulatory action. However, these studies adopted higher concentrations of CPZ than our current report, and sometime in non-immune cells. Therefore, further study will be highly warranted to identify which process(es) is impaired by $Psen2$ N141I mutation and how CPZ subdues overactivated PI3K/AKT1 pathway in microglia and other immune cells.

An interesting question is regarding the relative contributions of REV-ERBα's circadian or immunomodulatory effects to the phenotypes. Importantly, $PER2::Luc$ bioluminescence recordings from KI/+ microglia and BMDM confirmed that these cells remain rhythmic, although their amplitudes were decreased (Fig. 5e, f). Therefore, we propose that the hyperimmune response of a subset of cytokines was not due to a general disruption of the cellular clockwork, but REV-ERBα's diminished immunomodulatory activity.

Overall, our findings show that the $Psen2$ N141I mutation leads to dysregulated innate immunity through the epigenetic suppression of REV-ERBα and overproduction of a selective subset of clock gene-regulated cytokines and, mutant mice are highly vulnerable to immune challenge and predisposed to cognitive decline and AD immunopathology.

## Methods

**Animals**. All protocols and ethical regulations for the care and use of laboratory animals were approved by and in accordance with the guidelines established by the Institutional Animal Care and Use Committee of DGIST (DGIST-IACUC-210518 04-0001). Animals were maintained in a specific pathogen-free environment under a standard 12-h light/12-h dark cycle at the DGIST animal facility. $Psen2^{N141I/+}$ mice were generated using homologous recombination from C57BL/6J. Targeting vector included the I141 mutation in exon 4 and the $Neo^{r}$-$loxp$ sequence, and mutated sequence in the targeting vector was inserted into $Psen2$ of the WT allele. $Psen2^{N141I/N141I}$;$loxp$-$Neo^{r}$-$loxp$ mice were crossed with Cre mice to generate KI mice carrying the $Psen2$ N141I mutation using the Cre-loxp system. $Per2::Luc$ KI mice were a generous gift from Joseph Takahashi[37]. $Psen2^{N141I/+}$ mice were crossed with $Per2::Luc$ mice to monitor real-time circadian dynamics using bioluminescence recording. In all experiments, 8-week-old male mice were used and the number of mice in each experiment is provided in the figure legend.

**Reagents and antibodies**. The following antibodies were used: horseradish peroxidase–conjugated β-Actin (sc47778; 1:5000) from Santa Cruz Biotechnology; PSEN1 (5643; 1:1000), REV-ERBα (13418; 1:1000), DNMT3A (3598; 1:1000), AKT1 (2938; 1:2000), phosphorylated AKT1 (S473, 9271; 1:1000), MAPK1/3 (4969; 1:1000), phosphorylated MAPK1/3 (T203/Y205, 4370; 1:2000), GSK3β (9315; 1:1000) and phosphorylated GSK3β (9336; 1:1000) from Cell Signaling Technology; IBA-1 (019-19741; 1:250) from Wako Chemicals; N-cadherin (610920; 1:1000) from BD Bioscience; PSEN2 (ab51249; 1:2000) from Abcam; REV-ERBα (PA5-29865; 1:1000) from Thermo Fisher Scientific. The following reagents were used: LPS from $E.$ $coli$ 0111:B4 (L4391), DEX (50-02-2), LY294002 (L9908), DIF-3 (D0567) and chlorpromazine hydrochloride (C8138) were purchased from Sigma-Aldrich; 5-azacytidine (ab142744) from Abcam; and D-luciferin (E1601) from Promega; PD98059 (9900) from Cell Signaling Technology.

**Cell culture**. Primary microglia were obtained from 1–3-day-old neonatal mice brain with trypsinization and cultured in Dulbecco's modified Eagle's medium (DMEM, Corning) supplemented with 10% heat-inactivated fetal bovine serum (HI-FBS, Hyclone) and 1% penicillin–streptomycin (Hyclone). Microglia were isolated at days in vitro 12 by tapping. Purity of primary microglia and no contamination by astrocytes were confirmed by immunostaining with antibodies against IBA-1 and GFAP, specific microglia and astrocyte markers, respectively[68]. BMDM were obtained from femurs and tibias of 6–7-week-old mice with flushing to extrude bone marrow and were differentiated and grown in RPMI 1640 (GIBCO) medium supplemented with 10% HI-FBS and 1% penicillin–streptomycin[69].

MEF cells were obtained from pregnant female mice at gestational day 14.5–15.5. Embryos were harvested from the uterus and the head and visible internal organs were removed. Next, the tissue was rinsed with PBS, minced using a razor blade in 0.25% Trypsin-EDTA (T/E) and incubated for 25 min with pipetting intermittently for dissociation. Suspended cells were pelleted at 1000 × g for 3 min and collected cells were cultured in the MEF medium containing DMEM supplemented with 10% HI-FBS, 1% penicillin–streptomycin (Hyclone), and 2 mM L-glutamine for more than 3 days.

**Acute isolation of primary microglia and peritoneal macrophages**. The protocol was modified from the previous study[69]. PBS-perfused brains were extracted from 8-week-old mice and washed with 1× HBSS, chopped by surgical scissors and dissociated with tissue dissociation buffer (0.05% collagenase D, 5 mM $CaCl_2$, 20 U/mL DNase I, 15 mM HEPES and 0.5% glucose in HBSS) for 15 min at 37 °C. The dissociated tissues were passed through 70-µm pore mesh and pelleted at 500 × g for 3 min. Then, microglia cells were isolated using Percoll (P4937, Sigma-Aldrich) gradient (70%, 37%, and 30% Percoll in HBSS). The interphase between 70% and 37% Percoll was transferred to a new tube and centrifuged for 5 min at 500 × g. Each pellet was resuspended in MACS buffer, transferred to a new microcentrifuge tube, and microglia were isolated using CD11b+ microbeads (130-093-634,

Miltenyi Biotec). Peritoneal macrophages were obtained from peritoneum of 8-week-old C57BL/6 mice. Four days after i.p. injection of 3% Brewer thioglycolate medium[70], the needle filled with cold PBS was inserted through peritoneal wall into each anesthetized mice and peritoneal fluid was withdrawn slowly. Cell pellets were harvested after centrifugation.

**Quantitative RT-PCR**. RNA was isolated using the ImProm-II Reverse Transcriptase kit (Promega) and cDNA was synthesized using oligo dT. PCR primers were commercially synthesized (Cosmo Genetech). qRT-PCR was performed on the reverse transcribed product using Taq Polymerase (Invitrogen) and primers (Supplementary Table 1) specific for mouse cDNAs. TOPrealTM qPCR 2 × Pre-MIX (SYBR Green with low ROX) (Enzynomics) was used. Amplification (50 cycles) was performed in a CFX96 Real-Time System (Bio-Rad). *Actb* or 18S rRNA was used as the reference genes for normalization. CFX-manager (Bio-Rad) was used for analysis. The primers used are listed in Supplementary Table 1.

**Enzyme-linked immunosorbent assay**. ELISA kits for mouse IL-6, TNFα, IL-1β, CCL2, CXCL1, CCL5, CCL3, CCL4, and CXCL2, and a Proteome Profiler Mouse Cytokine Array Panel A kit were purchased from R&D Systems and used to measure cytokines in culture media and blood serum according to the manufacturer's instructions. Synergy HTX (Agilent Technologies) reader was used for the measurement of absorbance with Gen5 software (version 2.04) for data collection and analyses.

**Western blot analysis**. Cells were lysed in lysis buffer (1% Triton X-100, 250 mM sucrose, 1 mM EDTA, 1 mM phenylmethylsulfonyl fluoride, and 50 mM NaCl in 20 mM Tris-HCl, pH 7.4) with 1× protease and phosphatase inhibitors (Thermo Fisher Scientific) and 0.1 M dithiothreitol (Sigma-Aldrich). For REV-ERBα detection, cells were lysed in radio-immunoprecipitation assay buffer with 1× protease and phosphatase inhibitors and without 1 mM phenylmethylsulfonyl fluoride and 0.1 M dithiothreitol. Cell lysates were separated by SDS–polyacrylamide gel electrophoresis and transferred to polyvinylidene fluoride membranes. Membranes were incubated with the appropriate primary antibodies, and bound antibodies were detected by species-specific, horseradish peroxidase–conjugated secondary antibodies (1:5000). Chemiluminescence detection was performed to analyze the protein bands of interest. The blots were quantified using Image Studio lite 4.0 (LI-COR Biosciences).

**ChIP-qPCR analysis**. Cells were fixed and proteins were cross-linked to DNA using 0.75% paraformaldehyde (PFA) in PBS for 30 min. Glycine (125 mM) was added to quench cross-linking reaction, and cells were collected and lysed in radio-immunoprecipitation assay buffer (Sigma-Aldrich) containing 1× protease and phosphatase inhibitor cocktail, and DNA was fragmented by sonication to 500–1000-bp size. Protein A-agarose/salmon sperm DNA beads (Millipore) were incubated with DNA and antibody against the protein of interest or IgG, and the samples were immunoprecipitated overnight with rotation at 4 °C. DNA was eluted by adding elution buffer (10 mM EDTA, 1% SDS, 50 mM Tris-HCl, pH 8.0). Input samples were incubated with RNase A (20 μg/mL) and Proteinase K (0.5 mg/mL) at 65 °C overnight to reverse cross-linking. DNA was precipitated with isopropanol, and qRT-PCR was performed to assess the binding of the protein to the anticipated binding sites in DNA using the primers used listed in Supplementary Table 1.

**Y-maze assay**. Y-maze was used to evaluate spatial working memory. The assay was conducted in white plastic arms of a Y-shaped maze. A mouse was placed in the center and was allowed to freely explore the arms for 5 min. The experiment was recorded with EthoVision software 11.5 (Noldus). The number of arm entries and the number of triads were analyzed to calculate the percentage of alternation by dividing the number of three consecutive arm entries (triads) by the number of possible triads × 100 (total arm entries – 2).

**T-maze assay**. T-maze was used to evaluate spatial learning and memory with reward alternation. The assay was conducted in white plastic arms of a T-shaped maze. Mice were familiarized with the maze and food reward for 5 min before the test. Then, in the test run, rewards were placed in both arms, and one arm was blocked. Mice were placed at the base and ran to open arms to eat the reward. At the next trial, the previously closed arm was opened. Mice were placed back again at the base and chose one arm. If mice chose the newly opened arm, they were able to eat the reward. If mice incorrectly chose the previously visited arm, they did not get any rewards. The number of trials in which the correct arm was visited was expressed as a percentage of total arm entries. The experiment was recorded with EthoVision software 11.5 (Noldus).

**LABORAS**. Mice were placed in every single cage and recorded for 24 h starting from the night cycle at 7:00 PM. Activities of mice including locomotion, climbing, rearing, grooming, and eating were recorded and automatically analyzed using the Laboratory Animal Behavior Observation Registration and Analysis System (LABORAS, Metris).

**Microglial transduction**. sh*Nr1d1* (22747) was purchased from Addgene. Human *NR1D1* cDNA was cloned into the PLJM1-EGFP vector to generate lentiviruses. Lentiviruses were produced by transfection of Lenti-X 293T cells (632180, Clontech) with the transfer vector (PLKO.1-sh*Nr1d1* or PLJM1-*NR1D1*-EGFP), packaging vector (psPAX2, Addgene), and VSV-G envelope-expressing vector (PMD2.G, Addgene). After 3 days, the supernatant was harvested and ultra-centrifuged at 25,000 × g for 2 h (Optima XPN-100, Beckman Coulter) for virus concentration. Primary microglia were infected with the lentiviruses in a medium supplemented with hexadimethrine bromide (8 μg/mL). After 24 h, the medium was replaced with a fresh culture medium. eGFP expression was monitored using fluorescence microscopy after 72 h to estimate the lentiviral transduction efficiency.

**Immunohistochemistry**. Mice were anesthetized by injection of a mixture of Zoletil (Virbac, 50 mg/kg) and Rompun (Bayer, 10 mg/kg). Then, the mice were perfused with PBS, followed by 4% PFA. Brains were collected, post-fixed in 4% PFA for 16 h, transferred to 30% sucrose until they sank to the bottom of the tube, and frozen with optimal cutting temperature compound. Embedded brains were cut into 50-μm-thick coronal sections and slices were floated in PBS. Samples were incubated in sodium citrate buffer (10 mM tri-sodium citrate dihydrate, 1 N HCl, 0.05% Tween-20 in water) in a 1.5-mL tube at 95 °C for 5 min for antigen retrieval. Samples were washed with PBS three times and blocked with PBS containing 5% normal donkey serum and 0.01% Triton X-100. Slices were incubated with IBA-1 antibody in PBS containing 3% bovine serum albumin for 24 h at 4 °C and then with an appropriate secondary antibody for 2 h at room temperature. Images were acquired with LSM 7 and LSM 700 confocal laser scanning microscope and analyzed with ZEN 2010 software (Carl Zeiss).

**Stereotaxic injection**. FITC-conjugated Aβ$_{1-42}$ (4033502.05, BACHEM) was fibrilized in DMEM supplemented with 5% HI-FBS by overnight incubation at 37 °C[71]. Mice were anesthetized with Zoletil and Rumpun, and stereotaxic surgery was performed for 1 min with i.c.v. injection of 2.5 μL of fAβ42 (1.25 μM) in the right lateral ventricle at the speed of 0.25 μL/min using 10 μL Hamilton syringe and 27G needle. Coordinates were: AP –0.82 mm, ML +0.34 mm, DV –2.0 mm from the bregma.

**Microglia morphology analysis**. Confocal images were obtained along the entire Z-axis of a randomly selected field. Then, 3D images were reconstructed from confocal images using IMARIS software (version 9.2.1, Bitplane AG). Dendrite length, dendrite branch points, and Sholl radius of microglia were quantified using FilamentTracer function following IBA-1 signals.

**Bioluminescence recording**. To monitor real-time circadian dynamics in cells derived from *Per2::Luc* reporter mice, luminescence was continuously measured at 36 °C with a cell culture incubator–incorporated AB-2550 Kronos Dio luminometer (Atto) equipped with analysis software (version 2.10.233). Light emission was integrated for 1 min in 10-min intervals between measurements. Cells were synchronized with DEX (150 nM) in culture media. At 15 min after DEX addition, cells were washed and the medium was changed to a fresh recording medium (culture medium supplemented with 0.3 mM D-luciferin). Bioluminescence was analyzed in real time by the cosinor procedure[72,73].

**Methylation sequencing**. Genomic DNA was isolated from $10 \times 10^6$ microglial cells from WT and KI/+ mice using the MasterPure DNA purification kit (Epicentre) according to the manufacturer's instructions. The extracted genomic DNA was resuspended in TE buffer and quantified by fluorometry. Bisulphite conversion of genomic DNA was conducted using the EZ DNA Methylation Lightning kit (Zymo Research). In this method, non-methylated cytosine residues are converted into uracil by bisulphite treatment and read as thymine when sequenced. Methylated cytosines, protected from conversion, are still read as cytosines. Bisulphite-converted DNA was purified and used to prepare a sequencing library using the EpiGnome Methyl-Seq kit (Epicentre). In this procedure, to synthesize DNA containing a specific sequence tag, DNA is randomly primed using a polymerase able to read uracil nucleotides. Only the complement to the original bisulphite-treated DNA is used as the sequencing template; thus, the resulting Read 1 will always be the same sequence as the original bisulphite-treated strand.

**Methylation-specific PCR**. Genomic DNA was extracted from cells using genomic DNA extraction kit purchased from QIAGEN according to the manufacturer's instructions. Incubation of the genomic DNA with sodium bisulfite for conversion of all unmodified cytosine to uracil using bisulfite conversion kit purchased from New England Biolabs. PCR was performed on bisulfite-converted DNA using primers designed as in Supplementary Table 1. PCR conditions were as follows: 95 °C for 3 min; 34 cycles of 95 °C for 30 s the specific annealing temperature (unmethyl, 49 °C; methyl, 54 °C) for 30 s, and 72 °C for 1 min; and a final extension of 5 min at 72 °C.

**Statistical analysis**. Data were acquired in at least three independent experiments and are presented as mean ± standard error of the mean values. Statistical significance and *n* values are described in each figure legend. Statistical analysis was

performed by using Student's unpaired *t*-test, or one-way ANOVA and Tukey's post-test, or two-way ANOVA and Bonferroni post-test. Statistical significance was analyzed using GraphPad Prism 8.0.1 (GraphPad Software).

**Reporting summary**. Further information on research design is available in the Nature Research Reporting Summary linked to this article.

## Data availability

All the data supporting this study are available in the article and Supplementary information. The methylation sequencing data generated in this study have been deposited in NCBI Sequence Read Archive (SRA) PRJNA808254 (WT microglia: SRR18071758, SRR18071757, and SRR18071756; KI/+ microglia: SRR18071755, SRR18071754, and SRR18071753)[74]. Source Data are provided with this paper.

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

## Acknowledgements

We thank Dr Joseph Takahashi for providing *Per2::Luc* knock-in mice. This work was supported by the National Research Foundation of Korea (NRF) of the Ministry of Science and ICT of Korea through grants 2018M3C7A1056275 (to S.-W.Y., and E.-K.K.), 2020R1A2C4002156 (to E.-K.K.), and 2017R1C1B2008775 (to H.K.C.), and OATC Inc (S.-W.Y.). The funder had no role in the design, data collection and analyses, and publication of the study.

## Author contributions

S.-W.Y. designed the experiments, analyzed and interpreted the data, and wrote the manuscript. H.N. designed and performed most of the experiments, collected and analyzed the data, wrote the manuscript, and illustrated the schematic diagrams. Y.L., J.-W.L., Y.P., S.H., and B.K. performed additional experiments. B.K. and H.K.C. provided mice, and analyzed and interpreted the data. H.-K.A., K.M.C., and J.H. generated and provided *Psen2*^N141I/+ mice. K.K., E.-K.K., and H.K.C. provided critical experimental materials and revised the manuscript. All authors read and approved the final manuscript.

## Competing interests

S.-W.Y. and E.-K.K. are the scientific advisors to OATC Inc. S.-W.Y., H.N., and Y.L. have pending patent applications: international patent PCT/KR2021/015855 and Korean patent 10-2021-0150049. No potential conflict of interest was reported by all other authors.
