## [Peer Review File · Nature Communications]

Reviewers' comments:

Reviewer #1 (Remarks to the Author):

Review Nat Comm

In their paper, Nam and colleagues investigate the phenotype of mice heterozygous for a mutation in the presenilin 2 gene, which has been associated with Alzheimer disease in patients. This is generally a well-characterized study with multiple experimental readouts and applied techniques, yet the central mechanism is not known, namely how the mutation in Psen2 affects Rev-Erba expression.

Major concerns:

The authors observe a phenotype in the heterozygous mutants after inflammatory insults. However, what is the phenotype of mice and cells in non-inflammatory conditions?

The authors make a recurring point in mentioning clock-controlled cytokines and the effect of the mutation on their expression. In fact, this paper is not about the circadian clock, it appears to be a Rev-Erba-mediated phenomenon.

The synchronization phenotype in Figure 3 with respect to circadian oscillations observed in vitro appears not to hold true for the in vivo scenario. Since in these conditions there is no inflammation, are the authors saying that the observed reductions in amplitude are due to reduced expression of clock genes in cells or a reduced ability to synchronize? Or is there - in steady-state conditions - a pro-inflammatory phenotype that reduces amplitudes? This is an interesting phenomenon - given also the normal SCN oscillations - but it is currently unclear.

Lines 219-220: To prove that the mutation affects immune cells, the authors should do synchronization experiments with other cell types from these mice, e.g. fibroblasts etc. If these still exhibit a phenotype then it is likely more due to synchronization problems ex vivo.

The interactions between Psen2 and Rev-Erba is completely unclear. The authors should aim to uncover some of the mechanism linking these two molecules. While the methylation phenotype of Nr1d1 is very interesting, how this is linked to Psen2 is not known. Also, the mechanism of actions of CPZ in this scenario - which is certainly highly interesting - is not clear.

Clearly, the Nr1d1 KO only partially phenocopies the mutant described in this manuscript. What about the behavioral tests. Do Nr1d1 KO show similar defects (after light inflammation)?

Minor comments

line 56: the gene name is ARNTL

line 95: please explain FAD

some negative data should be outsourced to supplementary files (e.g. Figure 4)

Figure 2b: why are CCL3, CCL4 and CXCL2 downregulated in the het?

Reviewer #2 (Remarks to the Author):

The current manuscript provides a link between the Psen2 N141I FAD mutation and dysregulated innate immune response. They provide mechanistic insights by showing that the Psen2 N141I mutation induces hypermethylation in the promoter region of Nr1d1, leading to lower expression of REV-ERB and increased IL-6 level. Interestingly, treatment with 5-Aza restored the levels of Nr1d1 transcript and REV-ERB protein. The exacerbated inflammation in Psen2N141I/+ mice might be relevant to the memory deficits upon exposure to low doses of LPS. Furthermore, the antipsychotic chlorpromazine restores REV-ERB expression, increases IL6 level and thereby prevents the hyperactive inflammation and memory deficits in LPS-treated Psen2N141I/+ mice. While this study establishes a novel mechanism linking FAD mutation to neuroinflammation, there remain some major concerns about the manuscript in its current form that limit its impact for publication in NC.

1) Most of the study was carried out by using in vitro cultured microglia or BMDM, there's lack of in vivo evidences supporting the epigenetic suppression of REV-ERB α by Psen2 N141I mutation.

2) There's lack of mechanism by which Psen2 N141I mutation induces hypermethylation in the promoter region of Nr1d1. Furthermore, the changes in the Nr1d1-REV-ERB axis might not explain the increased expression of other circadian clock-controlled cytokines except IL6.

(3) It's unclear what assay is used for the compound screening. Thus, it remains to explain how chlorpromazine decreases the expression of clock-controlled cytokines, such as CXCL1, CCL2 and CCL5. Also what are the effects of chlorpromazine on microglial activation in Psen2 N141I mice.

(4) To establish a link to AD immunopathology, it's important to determine whether Abeta also induces hyperactive immune response in Psen2 N141I mice.

Reviewer #3 (Remarks to the Author):

In reference to paper submitted to Nature Comm entitled "Presenilin 2 N141I mutation induces hyperactive immune response 1 through the epigenetic repression of REV-ERB α " by Nam et al., my considerations after carefully read are as follow

The work is interesting, edge cutting on approach, and as a basic science impressive.

The fact that Psen2 mutations induced activation of immune system is known (as the authors indicate in their work) in fact the authors used this point to choose the best inflammatory candidate to follow up the genetic and pharmacological interventions they use to demonstrate hypothesis

As following it is also known the participation of epigenetic repression of REV-ERB α by repressing its gene expression of its gene, and this relationship with circadian rhythms

But certainly, is a new mechanism between Psen2 mutations and the alteration of immune system (mainly inflammatory pathway) and circadian cell alterations, the impact of this in the progression of AD or future treatment must be softened in the discussion.

The proposed use of chlorpromazine that is an antipsychotic with severe toxic effects that is indicated for psychiatric symptoms only when other safer APS drugs are not responsive is so much risky

The main concern is about the real impact on new therapeutic approaches for AD treatment as authors claims. This is an interesting mechanism of interactions among presenilin 2 mutations (that

constitute only a 5% of AD cases) and a transcription factor REV-ERB α , then really does not seem a new target or therapeutic approach to the AD treatment.

Methodology

IL-6, CXCL1, CCL2, and CCL5, are modified by KI Psen2 mice about WT, and IL1 β or TNF increased after LPS, both in KI and WT cell type analyzed

but only IL6 increased in front of an inflammatory stimuli.

There is a gap between the interest for Nr1d1 gene expression changes and other putative genes that can regulate cytokine levels and related with circadian cycle, that seems to be the main finding in the work. Authors focus on REV-ERB α codifying gene because Nr1d1 gene was increased and based on bibliography this transcription factor is that which match better with changes in IL6 and Tnf α . In some cases, in other tissues and obviously different models (some of them not related with AD or nervous cells). But they found also changes in Per 1 and 2 or Cry that they refused based on other works that are performed in other systems.

However, an experimental evidence is provided showing that Psen2 mutation did not change the expression/levels of other candidates than REV-ERB α . In some cases, the modification in expression were very significative (Fig 3). Authors can explain then why REV-ERB α was selected and the levels of the

Moreover, in demethylation experiment, only results for were showed, it is possible into the same samples to show the methylation pattern for other circadian genes related with inflammation and then discard with stronger evidence the participation of other circadian genes otherwise to confirm that all intern circadian cell system is altered under Psen 2 mutation rather than only a unique transcription factor.

An obvious epigenetic change is observed, and methylation patterns for the transcription factor selected are showed,

Can authors discard that Psen 2 mutation is altering methylation or demethylation machinery, (DNMT or TET enzymes) rather than affect a one transcription factor?

A few other genes/pathways can be altered after Psen 2 that will explain the changes in cognition in in vivo experiments.

In fact, the pharmacological strategies did not act on REV-ERB α action but on a plethora of other genes.

In my opinion with the bioinformatic tools used by the researchers it can be proposed other CpG islands methylation changes that must to be described in detail, and in this way authors can strength their conclusions about the importance of Psen2 mutation in the epigenetic control of genome and its implication in AD pathology development

Line 325, no changes about Nr1d1 gene expression in WT cells, but other gene expression? Also for KI Psen2 it could be of interest to study changes in the expression of other genes that are implicated in IL6 changes in immunity system cells (central and peripheric)

Rebuttal letter

Reviewer 1

1. The authors observe a phenotype in the heterozygous mutants after inflammatory insults. However, what is the phenotype of mice and cells in non-inflammatory conditions?

Response:

In the absence of inflammatory challenge, mutant mice are normal with intact memory ability. Their circadian rhythm and regular homecage behaviors are the same as WT mice. Microglia and BMDM derived from mutant mice remain rhythmic albeit with reduced amplitude, and do not produce cytokines in the absence of LPS treatment. Of interest, microglia in the hippocampus of mutant mice show hypertrophic, pro-inflammatory morphology at the basal state and chlorpromazine (CPZ) rescues this morphological change. **We added new data for the rescue of microglial morphology by CPZ at the basal state as well as after LPS stimulation *in vivo* in Fig. 9 and Supplementary Fig. 8.** Relevant description of the results was added from page 14 to page 15 in the Results.

2. The authors make a recurring point in mentioning clock-controlled cytokines and the effect of the mutation on their expression. In fact, this paper is not about the circadian clock, it appears to be a Rev-Erba-mediated phenomenon.

Response:

We agree with the reviewer and shortened the general introduction of circadian rhythm in the Introduction. We also revised the text and tried to focus on REV-ERB α -mediated immunomodulation. We changed "clock-controlled cytokines" to "clock gene-controlled cytokines".

3. The synchronization phenotype in Figure 3 with respect to circadian oscillations observed *in vitro* appears not to hold true for the *in vivo* scenario. Since in these conditions there is no inflammation, are the authors saying that the observed reductions in amplitude are due to reduced expression of clock genes in cells or a reduced ability to synchronize? Or is there - in steady-state conditions - a pro-inflammatory phenotype that reduces amplitudes? This is an interesting phenomenon - given also the normal SCN oscillations - but it is currently unclear.

Response:

CPZ rescued *Nr1d1* level, but not amplitude in KI/+ microglia. Therefore, we think that reduction in amplitude may be related with reduced ability to synchronize. We did not include this data (no change in circadian amplitude by CPZ) in the revised manuscript to stay focused on immunomodulation by REV-ERB α .

4. To prove that the mutation affects immune cells, the authors should do

synchronization experiments with other cell types from these mice, e.g. fibroblasts etc. If these still exhibit a phenotype than it is likely more due to synchronization problems ex vivo.

Response:

As suggested, we cultured MEFs from WT and KI/+ mice, and measured daily rhythm after synchronization. Interestingly, unlike reduced expression of REV-ERB α in microglia and BMDM, REV-ERB α protein level in mutant MEF cells were the same as WT and showed the same period and amplitude. Therefore, these data suggest that *Psen2* N141I mutation affects specifically immune cells. **We added these new data in the Supplementary Fig. 4.** Relevant description of the results was added in the page 11 of the Results.

5. The interactions between *Psen2* and Rev-Erba is completely unclear. The authors should aim to uncover some of the mechanism linking these two molecules. While the methylation phenotype of *Nr1d1* is very interesting, how this is linked to *Psen2* is not known. Also, the mechanism of actions of CPZ in this scenario - which is certainly highly interesting - is not clear.

Response:

As suggested, we investigated the underlying signaling mechanism and found that PI3K/AKT pathway is over-activated in mutant microglia and causes up-regulation of methylation enzymes (DNMT3A and DNMT3B) and down-regulation of demethylating TET1 enzyme (**a new figure as Fig. 10b, e, g-h**). DNMT3A, 3B and TET1 gene expression were also up-regulated and down-regulated in acutely isolated microglia and peritoneal macrophages isolated from mutant mic, respectively (**a new figure as Supplementary Fig. 9**). Relevant description of the results was added in the page 15 of the Results. CPZ inhibited PI3K/AKT, and decreased DNMT3A and 3B in mutant microglia (**a new figure as Fig. 10c-f**). To verify demethylation of *Nr1d1* by CPZ treatment, we performed methylation-specific PCR. 5-Aza (DNMT inhibitor) and CPZ treatment efficiently induced demethylation of hypermethylated *Nr1d1* promoter regions (**a new figure as Fig. 10a**). These data suggest that *Psen2* mutation causes activation of PI3K/AKT pathway, leading to up-regulation of DNA methylation enzymes, and CPZ demethylates *Nr1d1* gene by counteracting DNMT3A through the inhibition of PI3K/AKT pathway. **We added these new data in the figure 10** and relevant description of the results was added from page 15 to page 16 in the Results.

6. Clearly, the *Nr1d1* KO only partially phenocopies the mutant described in this manuscript. What about the behavioral tests. Do *Nr1d1* KO show similar defects (after light inflammation)?

Response:

Nr1d1 KO mice show altered mood-related behaviors and memory deficits at the basal state. However, *Psen2* N141I KI mutant mice show normal behaviors and memory in the absence of immune challenge. In terms of neuroinflammation, *Nr1d1* KO mice exhibit activated microglia morphology at baseline and elevated secretion of clock gene-controlled cytokines following LPS treatment. So, *Psen2* N141I KI mutant and *Nr1d1*

KO mice share proinflammatory phenotypes. Previous studies used much higher doses of LPS (1~2 mg/kg) with *Nr1d1* KO mice than our study (0.35 ug/kg). So, it remains to show whether *Nr1d1* KO mice will show the similar response as *Psen2* N141I KI mice after light inflammatory stimulation.

7. Minor comments:

7-1) line 56: the gene name is ARNTL

Response:

We correct this mistake and changed ARNTL1 to ARNTL.

7-2) Line 95: please explain FAD

Response:

FAD was defined in the first paragraph of the Introduction, page 3.

7-3) some negative data should be outsourced to supplementary files (e.g. Figure 4)

Response:

To concentrate on the immunomodulatory roles of REV-ERB α , we removed this figure and circadian oscillations of SCN culture.

7-4) Why are CCL3, CCL4 and CXCL2 downregulated in the het?

Response:

Although their secretion by LPS was reduced in the primary KI/+ microglia culture, their blood levels in LPS-injected KI/+ mice were not different from those of WT mice. These data suggest that reduction in their secretion seems to be limited to the in vitro microglia culture and there may be additional regulatory mechanisms in vivo.

Reviewer 2

1. Most of the study was carried out by using in vitro cultured microglia or BMDM, there's lack of in vivo evidences supporting the epigenetic suppression of REV-ERB α by *Psen2* N141I mutation.

Response:

We acutely isolated adult microglia and peritoneal macrophages from adult WT and mutant mice, and found the same reduction of *Nr1d1* expression and altered levels of DNA methylation machineries (DNMT3A/B and TET1) as in vitro cultures. **We added these new data in the Fig. 5d and Supplementary Fig. 9.** Relevant description of the results was added in the page 10 and 15 of the Results.

2. There's lack of mechanism by which *Psen2* N141I mutation induces hypermethylation in the promoter region of *Nr1d1*. Furthermore, the changes in the

Nr1d1-REV-ERB axis might not explain the increased expression of other circadian clock-controlled cytokines except IL6.

Response:

As explained in the Reviewer 1's comment, we found that overactivated PI3/AKT pathway unbalances DNA methylation machineries towards more methylation (Fig. 10). We also confirmed that REV-ERB α axis regulates the expression of other circadian gene-regulated cytokines (Supplementary Fig. 5). Relevant description of the results was added in the page 12 and from page 15 to page 16 in the Results.

3. It's unclear what assay is used for the compound screening. Thus, it remains to explain how chlorpromazine decreases the expression of clock-controlled cytokines, such as CXCL1, CCL2 and CCL5. Also what are the effects of chlorpromazine on microglial activation in *Psen2* N141I mice.

Response:

We screened the library by measuring IL-6 release, and filtered the hits on the basis of cytotoxicity, BBB penetration, and previously unknown anti-AD effects. We added screening scheme with the description of screening steps in Supplementary Fig. 7a.

CPZ rescued the hyperactive pro-inflammatory phenotypes at the baseline, and prevented LPS-induced morphological changes in *Psen2* KI/+ mice. We added these new data in the Fig.9 and Supplementary Fig.8. Relevant description of the results was added from page 14 to page 15 in the Results.

4. To establish a link to AD immunopathology, it's important to determine whether Abeta also induces hyperactive immune response in *Psen2* N141I mice.

Response:

In a similar manner to LPS, fibrillar form of A β 42 induced hyperimmune response and impaired memory function only in KI/+ mice by producing more clock gene-controlled cytokines. A β 42-induced increases in clock controlled-cytokines was successfully prevented by CPZ in KI/+ microglia. We added these new data in the Fig. 4 and Fig. 8e. Relevant description of the results was added in the page 9 and 14 of the Results.

Reviewer 3

1. ...But certainly, is a new mechanism between *Psen2* mutations and the alteration of immune system (mainly inflammatory pathway) and circadian cell alterations, the impact of this in the progression of AD or future treatment must be softened in the discussion.

Response:

As suggested, we toned down our statement in the page 18 of the Discussion.

2. ...The proposed use of chlorpromazine that is an antipsychotic with severe toxic effects only when other safer APS drugs are not responsive is so much risky. The main concern is about the real impact on new therapeutic approaches for AD treatment as authors claims.

Response:

We understand the reviewer's concern with the toxicity of CPZ and its therapeutic impact. However, our dosage is much lower than previously used in other animal studies. We would like to propose more finetuning of the dosages of CPZ, and softened this statement, as marked in red in the page 18 of the Discussion.

3. ...There is a gap between the interest for *Nr1d1* gene expression changes and other putative genes that can regulate cytokine levels and related with circadian cycle. ...But they found also changes in *Per 1* and *2* or *Cry* that they refused based on other works that are performed in other systems.

Response:

Through methylation analyses of the promoter regions, we found that only *Nr1d1* promoter, but not promoters of other clock genes, was hypermethylated, as explained in the page 13. We added these new data in the Fig.7b. CPZ also did not rescue the expression of other clock genes. We mentioned this as “data not shown” in the page 14 of the Results.

4. Authors can explain then why REV-ERB α was selected. Moreover, in demethylation experiment, it is possible into the same samples to show the methylation pattern for other circadian genes related with inflammation and then discard with stronger evidence the participation of other circadian genes otherwise to confirm that all intern circadian cell system is altered under *Psen2* mutation rather than only a unique transcription factor.

Response:

As mentioned in the comment 3, we found that the promoters of other circadian genes were not hypermethylated and not rescued by CPZ (Fig. 7b and data not shown in the page 14).

5. An obvious epigenetic change is observed, and methylation patterns for the transcription factor selected are showed, can authors discard that *Psen2* mutation is altering methylation or demethylation machinery, (DNMT or TET enzymes) rather than affect a one transcription factor?

Response:

As explained in the Reviewer 1's comment 5 and Reviewer 2's comment 2, we observed altered expression of methylation and demethylation enzymes, and rescue of this alteration by CPZ. We added these new data in the Fig.10.

6.In my opinion with the bioinformatic tools used by the researchers, it can be

proposed other CpG islands methylation changes that must be described in detail, and in this way authors can strength their conclusions about the importance of Psen2 mutation in the epigenetic control of genome and its implication in AD pathology development.

Response:

As the reviewer 3 suggested, we performed analyses of the promoter regions of clock genes, and found that only *Nr1d1* promoter was hypermethylated. **We added these new data in the Fig. 7b** and clarified this point in the page 13 of the Results.

7. Line 325, no changes about *Nr1d1* gene expression in WT cells, but other gene expression? Also for KI *Psen2* it could be of interest to study changes in the expression of other genes that are implicated in IL-6 changes in immunity system cells (central and peripheric)

Response:

Neither 5-Aza nor CPZ affected the expressions of other clock genes in WT cells. It would be interesting to investigate the changes of other pathways implicated in IL-6. We are also considering the possibility of positive feedback loop of IL-6 autocrine signaling. But this may be beyond the scope of the current study and we will follow this direction in our follow-up study.

REVIEWER COMMENTS

Reviewer #1 (Remarks to the Author):

The authors have done a good job in addressing my comments.

Reviewer #3 (Remarks to the Author):

Authors improved substantially evidences correlating the modulation by a mutated presenilin on Nr1d1 gene expression and subsequently into the transcription factor REV-ERB α , which in turn is able to control the production and release of cytokines. They demonstrate that this deleterious effect can be reversed by clozapine, because AKT/PI3K signalling is higher in Kin mice, but the cause for this overactivation can be multiple and downstream the action of the mutation in presenilin 2 N1211

Therefore, remains unclear how the Psen2 N1411/+ is able to modify the expression of the REV-ERB α by increasing the promoter hypermethylation. In fact, the authors answered that this is suggested, but they must establish a credible string of events demonstrating that Psen2 mutation overactivated AKT/PI3K pathway. This is probably a converged pathway in several putative mechanisms of modulation that could implicate not only AKT/PI3K but modification in the activity of several factors/proteins that activate/inhibits AKT/PI3. For example, MAPK or GSK3 β are also increased after Kin mice, Could a GSK3 or a MAPK inhibitor reduce also higher immunomodulation response after LPS challenge? Both GSK3 and MAPK can regulate AKT/PI3 activity (in a loop of regulation in kinases scenario).

Authors should discuss not only the pathway modified by clozapine but enter in deep in which is the effect of Psen2 N1411 mutation in immunocompetent cells.

The cartoon included is useful in part, because the reader have to imagine what is the cellular condition in Psen2 N1411/+ mice after Clozapine.. perhaps will be more useful to include a third square drawing the restoring effect of Clozapine, but it is understandable that in this case the Psen2 N1411/+ mutation effect before the hypermethylation activity is not easy to explain

Reviewer #4 (Remarks to the Author):

The points 2-4 of referee #2 were well addressed. Especially, the experiments in response to point 4 are convincing as the injection of fibrillary Abeta 1-42 causes similar effects as LPS injection in the KI/+ mice, thus supporting the link to AD pathology.

Regarding #1, a more detailed epigenetic analysis of the freshly isolated microglia and macrophages would have been interesting, but the results presented in Fig. Supplementary Fig. 9 are sufficiently satisfying at this point.

Response letter

Reviewer 1

The authors have done a good job in addressing my comments.

Reviewer 3

Authors improved substantially evidences correlating the modulation by a mutated presenilin on *Nr1d1* gene expression and subsequently into the transcription factor REV-ERB α , which in turn is able to control the production and release of cytokines. They demonstrate that this deleterious effect can be reversed by clozapine, because AKT/PI3K signalling is higher in Kin mice, but the cause for this overactivation can be multiple and downstream the action of the mutation in presenilin 2 N1211.

Therefore, remains unclear how the *Psen2* N1411/+ is able to modify the expression of the REV-ERB α by increasing the promoter hypermethylation. In fact, the authors answered that this is suggested, but they must establish a credible string of events demonstrating that *Psen2* mutation overactivated AKT/PI3K pathway. This is probably a converged pathway in several putative mechanisms of modulation that could implicate not only AKT/PI3K but modification in the activity of several factors/proteins that activate/inhibits AKT/PI3. For example, MAPK or GSK3 β are also increased after Kin mice, Could a GSK3 or a MAPK inhibitor reduce also higher immunomodulation response after LPS challenge? Both GSK3 and MAPK can regulate AKT/PI3K activity (in a loop of regulation in kinases scenario).

Authors should discuss not only the pathway modified by clozapine but enter in deep in which is the effect of *Psen2* N1411 mutation in immunocompetent cells.

The cartoon included is useful in part, because the reader have to imagine what is the cellular condition in *Psen2* N1411/+ mice after Clozapine.. perhaps will be more useful to include a third square drawing the restoring effect of Clozapine, but it is understandable that in this case the *Psen2* N1411/+ mutation effect before the hypermethylation activity is not easy to explain

Response:

As the reviewer indicated, GSK3 β , MAPK, and PI3K/AKT1 can regulate each other in a loop of kinases signaling. So, we tested various doses of MAPK inhibitor (PD98059) and GSK3 β activator (DIF-3, GSK3 β activity is reduced in KI/+ microglia) to finetune the experimental condition and understand the roles of these kinases and their interaction.

PD98059 significantly reduced mRNA levels of *Il-6* and *Tnf* in both WT and KI/+ microglia in a dose-dependent manner, but did not affect PI3K/AKT1 or *Nr1d1* level at the tested doses. So, MAPK seems to regulate immune response independently of PI3K/AKT1 under our experimental condition.

DIF-3 at 5 μ M did not activate GSK3 β nor reduced inflammation. DIF at the concentration of 7.5 μ M was able to activate GSK3 β and reduced mRNA levels of *Il-6* and *Tnf* in both WT and KI/+ microglia, but failed to affect PI3K/AKT1 or rescue *Nr1d1* level. When we increased DIF-3 concentration, it was toxic to the cells.

Based on these data, we think that MAPK or GSK3 β is not intimately associated with PI3K/AKT1 overactivation. We added new data obtained with PD98059 and DIF-3 as a new figure (Supplementary Figure 10). Relevant descriptions of Results and Methods were added in page 16 and 21.

We agree with the reviewer that *Psen2* N141 I mutation has diverse cellular consequences and can activate PI3K/AKT by multiple mechanisms including other kinases. Indeed, we are very interested in this direction of study and investigating several processes. However, this manuscript already has many data to demonstrate hyper immune reactivity caused by *Psen2* mutation. We hope to be able to address this important question in our follow-up study.

Psen2 N141I mutation was known to affect various cellular processes, including autophagy, intracellular Ca²⁺, and oxidative stress response. We included the cellular pathways that *Psen2* mutation may affect and discussed how chlorpromazine may modify these processes to rescue PI3K/AKT overactivation in the page 19.

Reviewer 4

The points 2-4 of referee #2 were well addressed. Especially, the experiments in response to point 4 are convincing as the injection of fibrillary Abeta 1-42 causes similar effects as LPS injection in the KI/+ mice, thus supporting the link to AD pathology.

Regarding #1, a more detailed epigenetic analysis of the freshly isolated microglia and macrophages would have been interesting, but the results presented in Fig. Supplementary Fig. 9 are sufficiently satisfying at this point.

REVIEWERS' COMMENTS

Reviewer #3 (Remarks to the Author):

Authors included interesting new results, and improved significantly the work given new clues about the implication of Psen2 N1411/+ 1 mutation and changes in REV-ERB α epigenetic marks

This is enough to consider the work as acceptable for publication in Nat Com